# Regulating vesicle bilayer permeability and selectivity via stimuli-triggered polymersome-to-PICsome transition

Xiaorui Wang[1,2], Chenzhi Yao[1], Guoying Zhang[1] & Shiyong Liu [1✉]

Compared to liposomes, polymersomes of block copolymers (BCPs) possess enhanced stability, along with compromised bilayer permeability. Though polyion complex vesicles (PICsomes) from oppositely charged block polyelectrolytes possess semipermeable bilayers, they are unstable towards physiologically relevant ionic strength and temperature; moreover, permselectivity tuning of PICsomes has remained a challenge. Starting from a single component diblock or triblock precursor, we solve this dilemma by stimuli-triggered chemical reactions within pre-organized BCP vesicles, actuating in situ polymersome-to-PICsome transition and achieving molecular size-selective cargo release at tunable rates. UV light and reductive milieu were utilized to trigger carboxyl decaging and generate ion pairs within hydrophobic polymersome bilayers containing tertiary amines. Contrary to conventional PICsomes, in situ generated ones are highly stable towards extreme pH range (pH 2-12), ionic strength (~3 M NaCl), and elevated temperature (70 °C) due to multivalent ion-pair interactions at high local concentration and cooperative hydrogen bonding interactions of pre-organized carbamate linkages.

[1] CAS Key Laboratory of Soft Matter Chemistry, Hefei National Laboratory for Physical Sciences at the Microscale, Department of Polymer Science and Engineering, University of Science and Technology of China, 230026 Hefei, Anhui, China. [2] Guangdong Provincial Key Laboratory of Construction and Detection in Tissue Engineering, Biomaterials Research Center, School of Biomedical Engineering, Southern Medical University, 510515 Guangzhou, China. ✉email: sliu@ustc.edu.cn

Mimicking intricate structures and functions of complex biological systems such as cell membranes and viral capsids has inspired the creation of artificial nanos-tructures including liposomes and polymeric vesicles (poly-mersomes)[1–6], both consisting of aqueous interiors enclosed by hydrophobic bilayers[7–11]. Compared to liposomes self-assembled from small molecule lipids, polymersomes of amphiphilic block copolymers (BCPs) possess much improved microstructural stability. They have been increasingly utilized to fabricate delivery of nanovehicles[12–14], nanoreactors[15–18], and artificial organelles[3,19,20]. However, effective permeation of active agents through/from aqueous lumens is prohibited by the hydrophobicity of thick bilayer membranes[1,11]. To solve this issue, several approaches such as membrane integration with channel proteins, copolymerization with responsive moieties, and bilayer post-modification have been proposed[3,21–25]. These approaches typically involve tedious procedures and need to introduce external additives, leading to unsatisfactory perms-electivity and loss of structural integrity in certain cases. We previously proposed the concept of "traceless crosslinking" and achieved concurrent vesicle crosslinking and membrane per-meabilization[26–28]. However, the nature of irreversible bilayer chemical crosslinking restricts their in vivo applications con-sidering biodegradation and clearance issues; in addition, it still remains a challenge to achieve molecular size-selective release and permselectivity regulation features.

In contrast to polymersomes with hydrophobic bilayers, polyion complex vesicles (PICsomes) constructed from two oppositely charged block polyelectrolytes possess semipermeable bilayers and improved permeability towards hydrophilic solutes[29–31]. However, PICsomes are intrinsically unstable under physiologically relevant temperature and ionic strength due to dynamic exchange with unit PICs, and their use as in vivo nanocarriers also need covalent stabilization via chemical crosslinking[30,32–36]. Though the stability issue has recently been partially solved by strategies, such as introducing guanidinium hydrogen bonding (HB) motif[37] and longer alkyl spacer[38] into the cationic block, and use of strong block polyelectrolyte[39,40] or polyions with a dendritic topology[41,42], PICsomes still encounter several major limitations. First, its fabrication process is not compatible with hydrophobic drugs and imaging agents[29]. Sec-ond, the formation of PICsomes involves two oppositely charged components with at least one of them being block polyelec-trolytes[29–31]; the possibility of PICsome formation from a single component with oppositely charged comonomers arranged in a random, block, or alternating manner (e.g., block poly-ampholytes) remains to be explored[43,44]. Finally, the permselec-tivity modulation of PICsome bilayers has not been achieved yet, and macromolecular agents up to a molar mass of 10 kDa (e.g., dextran) could easily permeate though PICsome bilayers[30]. Thus, the loading capability and encapsulation stability of PICsomes towards hydrophilic small molecule drugs and functional agents need to be further improved.

Although both polyion complex (PIC) micelles[45–47] and PIC-somes[29–31] are intrinsically sensitive to high ionic strength, pH, and temperature[40], previous literature reports also hint, in a retrospective view, that the stability of PICs or inter-polyelectrolyte complexes (IPECs) is highly dependent upon local concentrations and sequence arrangement of charged ion-pairs[43,44,48–51]. Sun et al.[48] fabricated tough and viscoelastic polyamphoyte hydrogels via direct copolymerization of oppo-sitely charged comonomers at >1.5 M total concentration in aqueous media containing 0.5 M NaCl. The resultant supramo-lecular hydrogels are stable towards high salt concentrations and elevated temperatures. For model peptides containing a well-positioned single pair of histidine and aspartate partially buried in

an alanine-rich hydrophobic milieu, ion-pair interactions assisted by HB are not screened by high ionic strength up to ~1.0 M NaCl, contributing to helix stability[51]. Thus, rationally designed ion-pair interactions with preferred orientation and high density could provide Coulomb attraction instead of repulsion[52].

We then envisage that attractive ion-pair interactions could be exploited to stabilize nanostructures, such as PICsomes against physiologically relevant conditions if they are present at high local density and cooperatively assisted by other types of noncovalent interactions (e.g., HB, π–π, and hydrophobic). By taking advan-tage of the high local concentration of functional moieties (~1–2 mol/L) within hydrophobic bilayers of self-assembled polymersomes[52], we herein propose a general strategy to in situ generate ion-pair interactions and trigger transformation from polymersomes to PICsomes (Fig. 1).

Polymersomes were self-assembled from amphiphilic BCPs containing tertiary amine (DPA or DEA) and caged carboxyl

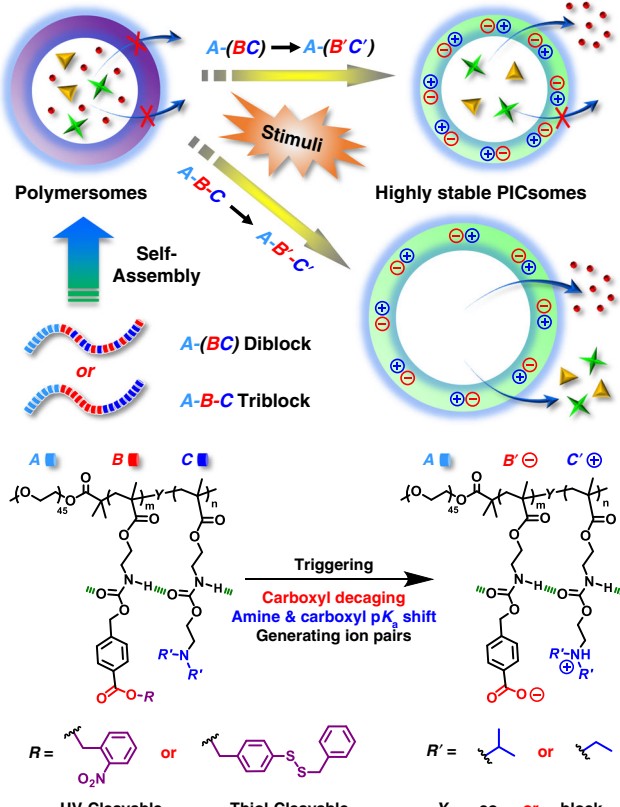

**Fig. 1 Stimuli-triggered polymersome-to-PICsome transition and concurrent permeability regulation.** For polymersomes self-assembled from amphiphilic diblock and triblock copolymers containing tertiary amine and caged carboxyl moieties in the hydrophobic block, external stimuli including UV light and reductive milieu trigger the decaging of carboxyl functionalities, which transfer protons onto neighboring tertiary amine moieties and render ion-pair interactions. Note that in situ generated PICsomes are highly stable towards extreme pH range, high ionic strength, and elevated temperature due to cooperative ion-pair interactions at high local concentration within pre-organized vesicle bilayers and synergistic contributions from hydrogen bonding interactions of carbamate side linkages. The polymersome-to-PICsome transition is accompanied with the transformation of hydrophobic bilayers into semipermeable membranes and switching of vesicle bilayer permeability. Moreover, sequence structure of the bilayer forming block, A-(B-co-C) vs. A-B-C, could be further utilized to regulate the permselectivity of resultant semipermeable PICsome bilayers.

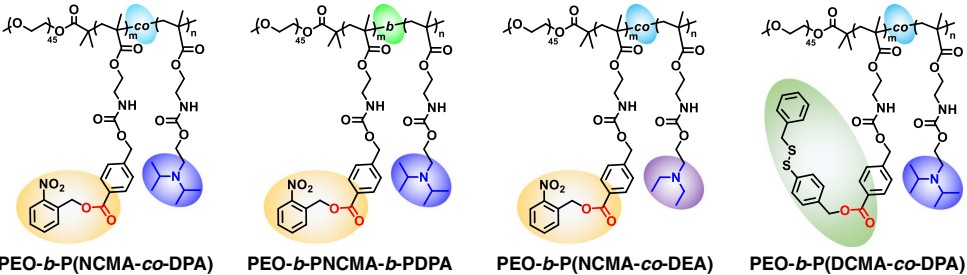

**Fig. 2** Chemical structures of four types of amphiphilic block copolymers (BCPs) used in this study.

comonomers (NCMA or DCMA) in the stimuli-responsive block, being initially hydrophobic (Fig. 2). We intentionally integrate carbamate linkage into both tertiary amine-containing comonomers and caged carboxyl comonomers to strengthen intra- and inter-chain HB interactions within vesicle bilayers[26–28,52], which provide a pre-organized milieu for cooperative noncovalent interactions with directional orientation and high local concentration. Upon triggered cleavage and generation of carboxyl moieties within bilayers, proton transfer from carboxyl to tertiary amine in situ generates ionized carboxyl and protonated amine moieties. Ion-pair interactions are strengthened by carbamate-relevant HB interactions and the hydrophobic bilayer-forming scaffold, contributing cooperatively to the high stability of in situ formed PICsomes towards extreme pH, high ionic strength, and elevated temperature.

The strategy of in situ-triggered polymersome-to-PICsome transition bridges two main types of vesicles assembled from BCPs, starting from a single component block polyampholyte precursor. This feature differs from that of conventional PICsomes, which involves two oppositely charged polyion components. We further demonstrate that both UV light and reductive milieu could trigger polymersome-to-PICsome transition accompanied with the switching of vesicle bilayer permeability, manifesting the generality of the proposed strategy. Moreover, the permselectivity of resultant PICsomes could be finely tuned by the sequence arrangement of polyions (random vs. block type), and molecular size-selective release from PICsome interiors could be successfully achieved (Fig. 1). Note that this feature has not been previously realized for conventional PICsomes.

## Results

**Synthesis and self-assembly of amphiphilic BCPs.** Aiming to fulfill the above design rationale, two series of carbamate-containing monomers possessing caged carboxyl and tertiary amine functionalities were designed (Supplementary Fig. 1). 2-Nitrobenzyl ester photo-caged carboxyl monomer (NCMA) and disulfide-caged carboxyl monomer (DCMA) generate carboxyl moieties upon actuation of UV irradiation and reductive milieu, respectively[26–28]. For the second series, pH-responsive DPA and DEA monomers containing both carbamate linkage and tertiary amine moieties were synthesized (Supplementary Fig. 1c)[53]. These four types of monomers (NCMA, DCMA, DPA, and DEA) and relevant synthetic intermediates were well-characterized by $^1$H and $^{13}$C NMR analysis (Supplementary Figs. 1 and 6–10). Next, reversible addition-fragmentation chain transfer (RAFT) polymerizations using PEG$_{45}$-based macro-RAFT agent afforded a series of amphiphilic BCPs with varying comonomer sequences and compositions, including PEO$_{45}$-b-P (NCMA$_{0.55}$-co-DPA$_{0.45}$)$_{29}$, PEO$_{45}$-b-P(NCMA$_{0.49}$-co-DEA$_{0.51}$)$_{32}$, and PEO$_{45}$-b-P(DCMA$_{0.45}$-co-DPA$_{0.55}$)$_{33}$ diblock copolymers, and PEO$_{45}$-b-PNCMA$_{17}$-b-PDPA$_{21}$ triblock copolymer (Fig. 2, and Supplementary Figs. 2 and 3). These BCPs were characterized by $^1$H NMR and GPC analyses (Supplementary Figs. 11–14)

and their structural parameters are summarized in Supplementary Table 1.

For the hydrophobic block in these diblock and triblock copolymers, an almost equal ratio of caged carboxyl and tertiary amine comonomers was chosen. Upon decaging, carboxyl and tertiary amine moieties will be at roughly equivalent molar ratio, thus facilitating cooperative ion-pair interactions within vesicle bilayers (Fig. 1)[14]. Additionally, two types dye-labeled amphiphilic BCPs, PEO$_{45}$-b-P(NCMA$_{0.55}$-co-DPA$_{0.45}$)$_{29}$-Nile red and PEO$_{45}$-b-P(NCMA$_{0.55}$-co-DPA$_{0.45}$)$_{29}$-naphthalimide were also synthesized to fabricate vesicles conjugated with microenvironmental polarity-sensitive and pH-sensitive fluorescent probes (Supplementary Fig. 4). Furthermore, two types of control amphiphilic BCPs, PEO$_{45}$-b-PNCMA$_{30}$ and PEO$_{45}$-b-PPA$_{26}$, were also synthesized (Supplementary Figs. 5 and 15). For as-synthesized diblock and triblock copolymers, tertiary amine comonomer units are located within hydrophobic microenvironment containing caged carboxyl moieties, which considerably suppresses the apparent p$K_a$ for tertiary amines. For example, the amine p$K_a$ was determined to be ~5.7 for PEO$_{45}$-b-P(NCMA$_{0.55}$-co-DPA$_{0.45}$)$_{29}$, in contrast to the apparent p$K_a$ of ~6.4 for PDPA homopolymer (Supplementary Fig. 16)[53].

BCP self-assembly was triggered by slow addition of water into the polymer solution in acetone. Transmission electron microscopy (TEM) observation revealed the presence of typical vesicular nanostructures for all four types of BCPs (Fig. 3a and Supplementary Fig. 17). Dynamic laser light scattering (DLS) analysis revealed that resultant polymersomes possess intensity-average hydrodynamic diameters, $\langle D_h \rangle$, in the range of 530–640 nm and relative low polydispersities ($\mu_2/\Gamma^2$ ~ 0.1), which are in general agreement with TEM results (Supplementary Table 1).

**Light-triggered polymersome-to-PICsome transition.** Next, we investigated UV light-triggered evolution of PEO$_{45}$-b-P (NCMA$_{0.55}$-co-DPA$_{0.45}$)$_{29}$ polymersomes in aqueous media. As photo-labile NCMA and DPA comonomers are in a random sequence within the hydrophobic block, they should be in close contact with each other within polymersome bilayers. Upon UV irradiation, the process of photocleavage of 2-nitrobenzyl units and generation of carboxyl moieties were monitored by time-dependent UV/Vis absorption spectra (Supplementary Fig. 18). The photocleavage occurred quickly within the initial ~5 min and then leveled off at ~10 min under 365 nm LED light irradiation. UV-triggered carboxyl decaging process was further examined with $^1$H NMR (Supplementary Fig. 19), demonstrating a photocleavage extent of >98% upon UV irradiation for 10 min. We also attempted to characterize UV-triggered chemical structural changes by matrix-assisted laser desorption/ionization time-of-flight mass spectrometry (MALDI-TOF MS) technique, but no reliable signals could be recorded for both non-irradiated and UV-irradiated polymersome dispersions upon lyophilization (Supplementary Fig. 20).

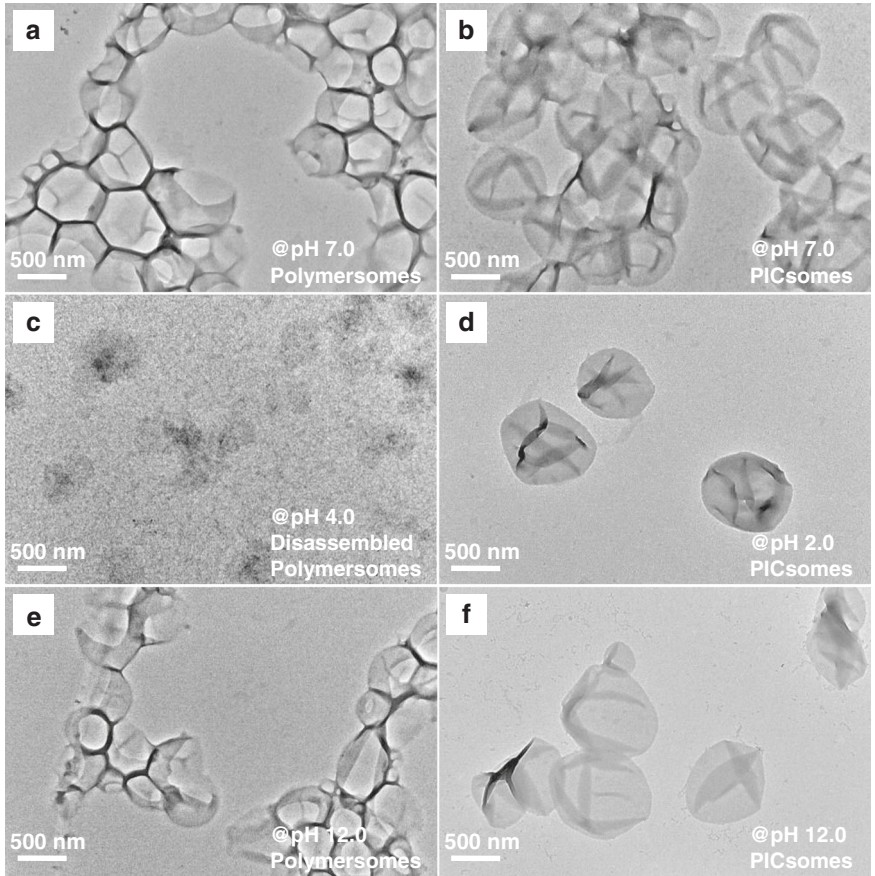

**Fig. 3 TEM images of PEO$_{45}$-*b*-P(NCMA$_{0.55}$-*co*-DPA$_{0.45}$)$_{29}$ polymersomes and corresponding PICsomes.** Original polymersomes **a** and PICsomes obtained after 10 min UV irradiation in neutral aqueous media **b**; **c**, **e** Polymersomes after being subjected to pH 4.0 and pH 12.0, exhibiting microstructural destruction in acidic media; **d**, **f** PICsomes in aqueous media at pH 2.0 and 12.0.

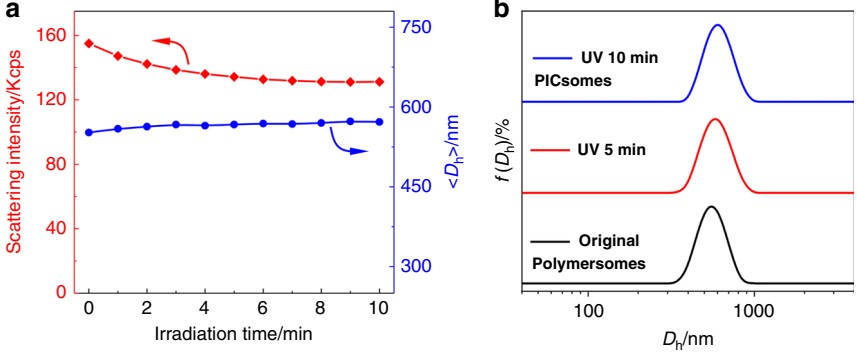

**Fig. 4 DLS characterization of polymersome-to-PICsome transition upon UV irradiation. a** Irradiation duration-dependent evolution of scattered light intensities and $\langle D_h \rangle$ recorded for PEO$_{45}$-*b*-P(NCMA$_{0.55}$-*co*-DPA$_{0.45}$)$_{29}$ vesicles in neutral aqueous media. **b** Intensity-average hydrodynamic diameter, $D_h$, distributions before and after UV irradiation.

Apparently, the vesicular dispersion initially exhibited a bluish tinge, but changed to grayish yellow after UV irradiation (insets in Supplementary Fig. 21). Furthermore, $^1$H NMR spectra of the vesicle dispersion in D$_2$O before and after UV irradiation showed only signals of well-solvated PEO coronas in both cases (Supplementary Fig. 21), indicating the presence of colloidal aggregates after UV-triggered carboxyl decaging. Moreover, direct TEM observations after UV irradiation revealed the presence of intact and robust vesicular nanostructures (Fig. 3b).

Time-dependent DLS measurements were conducted to track the evolution of scattered light intensities and intensity-average hydrodynamic diameter, $\langle D_h \rangle$, upon UV irradiation (Fig. 4a).

Scattered light intensities of PEO$_{45}$-*b*-P(NCMA$_{0.55}$-*co*-DPA$_{0.45}$)$_{29}$ polymersome dispersion exhibited an initial ~15% decrease within ~5 min UV irradiation and then reached a plateau, this was accompanied with a slight increase of $\langle D_h \rangle$ from 550 to 570 nm upon 10 min UV irradiation. In addition, the $D_h$ distribution remained almost unchanged upon UV irradiation (Fig. 4b). Moreover, if we replace the *N,N*-diisopropylamine motif in the above BCP with *N,N*-diethylamine, the resultant PEO$_{45}$-*b*-P(NCMA$_{0.49}$-*co*-DEA$_{0.51}$)$_{32}$ BCP (Fig. 2 and Supplementary Table 1) should possess higher apparent p$K_a$ for tertiary amines (Supplementary Fig. 16). Again, both DLS and TEM characterization results confirmed UV light-triggered polymersome-to-PICsome transition,

indicating the generality of the proposed strategy (Supplementary Fig. 22). To exclude possible UV-triggered photoreaction and/or decomposition of tertiary amine moieties in BCPs, we further examined the photostability of both *N,N*-diisopropylethylamine and triethylamine against UV irradiation, revealing essentially no discernible chemical structural changes (Supplementary Figs. 23 and 24).

The above results indicated that vesicles possess microstructural integrity after UV irradiation although light-triggered cleavage of 2-nitrobenzyl ester generated hydrophilic carboxyl moieties. Upon carboxyl generation, proton transfer from carboxyl to tertiary amine moieties will occur. This is reasonable considering that most of tertiary amines are initially in the unprotonated state (p$K_a$ ~ 5.7, Supplementary Fig. 16), and carboxyl functionalities of the control BCP, PEO$_{45}$-*b*-PPA$_{26}$, possess an apparent p$K_a$ of ~6.5 (Supplementary Fig. 25). Most importantly, the mutual presence of neighboring amine and carboxyl functionalities, and the generated hydrophilic milieu will considerably increase and decrease p$K_a$ values of amine and carboxyl moieties, respectively, thus facilitating proton transfer and ion-pair formation. Indeed, it is well-accepted that protonated amines tend to increase the acidity of neighboring

carboxylic acids by stabilizing the conjugate base (carboxylate) via electrostatic interactions. We tentatively propose that upon UV decaging, vesicular nanostructures are stabilized by newly generated ion-pair interactions at high local concentrations and synergistically enhanced by side chain HB interactions (Fig. 1 and Supplementary Fig. 26). Note that before UV irradiation, original polymersomes are mainly stabilized by hydrophobic interactions; whereas UV-irradiation in situ generates extensive ion-pair interactions within vesicle bilayers, thus corresponding to light-actuated polymersome-to-PICsome transition. For resultant PICsomes, potentiometric titration experiments revealed that ion-pairs form within the pH range of 4.6–8.8; whereas below pH 4.6 and above pH 8.8, ionized carboxylates and protonated amines did not exist within vesicle bilayers, respectively (Supplementary Fig. 27).

To probe the extent of proton transfer and ion-pair formation, we fabricated polymersomes conjugated with pH-sensitive fluorescent probe via the co-assembly of PEO$_{45}$-*b*-P(NCMA$_{0.55}$-*co*-DPA$_{0.45}$)$_{29}$ and PEO$_{45}$-*b*-P(NCMA$_{0.55}$-*co*-DPA$_{0.45}$)$_{29}$-*naphthalimide* BCPs (8:2 molar ratio) (Fig. 5a). Note that in the latter, the tertiary amine of naphthalimide probe possesses an apparent p$K_a$ comparable to those of PDPA homopolymer (Supplementary

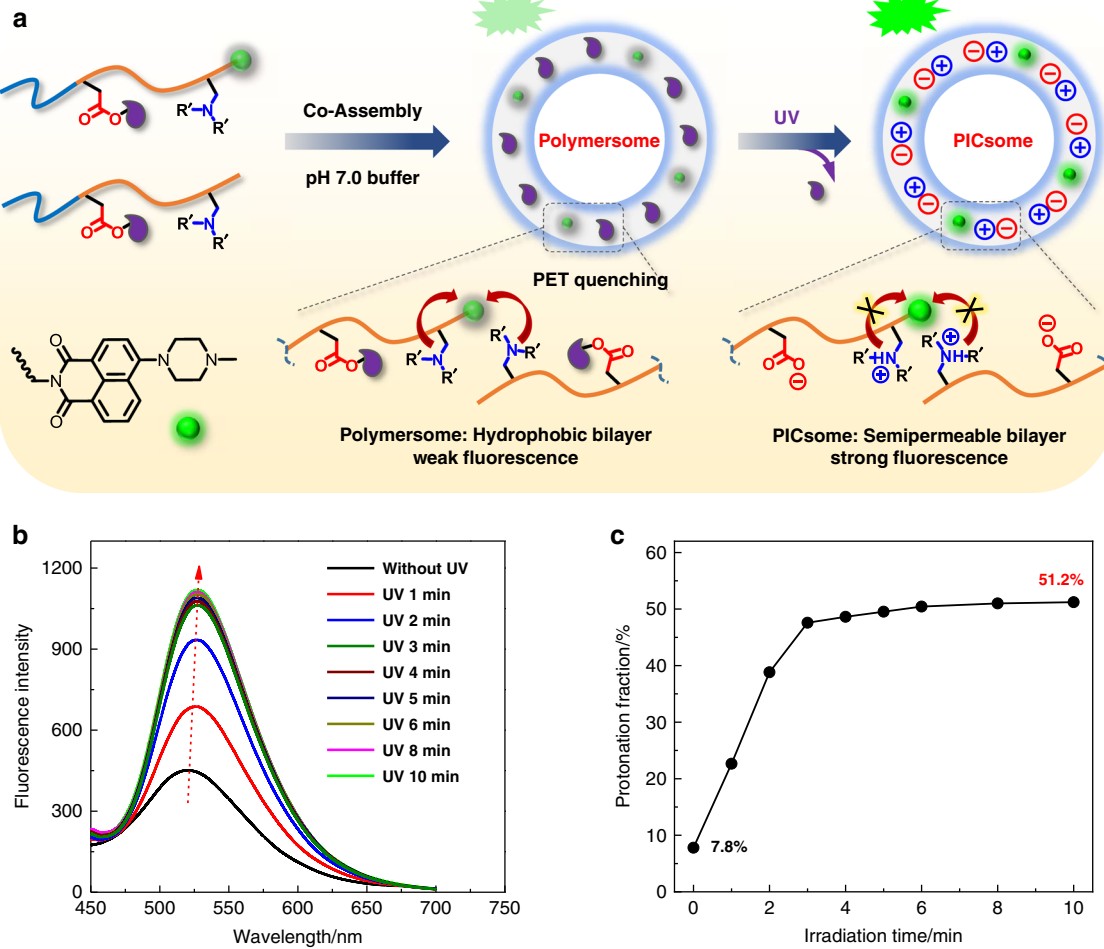

**Fig. 5 In situ proton transfer from newly generated carboxyl to amine species and formation of ion-pairs. a** Schematics of the fabrication of naphthalimide-labeled polymersomes co-assembled from PEO$_{45}$-*b*-P(NCMA$_{0.55}$-*co*-DPA$_{0.45}$)$_{29}$ and PEO$_{45}$-*b*-P(NCMA$_{0.55}$-*co*-DPA$_{0.45}$)$_{29}$-*naphthalimide* (8:2 wt/wt); initially, naphthalimide emission is partially quenched by neighboring tertiary amine moieties via PET mechanism; upon decaging of carboxyl moieties and protonation of amine moieties, naphthalimide emission is prominently enhanced. **b** Evolution of fluorescence emission spectra during polymersome-to-PICsome transition upon UV irradiation. **c** Irradiation time-dependent evolution of the protonation fraction of tertiary amine moieties within vesicle bilayers during polymersome-to-PICsome transition. All data were obtained at a polymer concentration of 0.1 g/L at [NCMA] ~ 0.12 mM and [DPA] ~ 0.10 mM in Britton–Robinson buffer (pH 7.0, 12 mM; 25 °C).

Fig. 16[54,55]. As shown in Supplementary Fig. 28, for naphthalimide-conjugated polymersome dispersion in aqueous media, decreasing solution pH from 7 to 4 led to emission enhancement due to the loss of PET quenching for protonated tertiary amines; whereas in the pH range of 7–9, fluorescence emission exhibited only a slight decrease, which is in agreement with the apparent amine p$K_a$ (~5.7) for PEO$_{45}$-$b$-P(NCMA$_{0.55}$-$co$-DPA$_{0.45}$)$_{29}$ (Supplementary Fig. 16). For PICsomes obtained via UV irradiation, pH-dependent transition range of emission intensities shifted to higher pH, and the degree of amine protonation was determined to be ~51.2% at pH 7 (Supplementary Fig. 29).

During UV light-triggered polymersome-to-PICsome transition under Britton–Robinson buffer media (pH 7.0, 12 mM; 25 °C), fluorescence emission intensities increased rapidly within the first 3–4 min (Fig. 5), which agrees with decaging kinetics shown in Supplementary Figs. 18 and 19. We further confirmed that upon UV light irradiation, naphthalimide dye exhibited negligible photobleaching (Supplementary Fig. 30). The above results clearly confirmed in situ proton transfer from newly generated carboxyl to amine species and formation of ion-pairs (Fig. 1). Moreover, on the basis of results shown in Supplementary Figs. 28 and 29 for dispersions of polymersomes and PICsomes before and after UV irradiation, respectively, the extent of amine protonation increased from ~7.8% to ~51.2% during polymersome-to-PICsome transition (Fig. 5c). This indicated that ~50% carboxyl and amine moieties formed ion-pairs within resultant PICsomes (Fig. 1).

For the control BCP, PEO$_{45}$-$b$-PNCMA$_{30}$, its polymersome dispersion in neutral aqueous media exhibited an apparent pH decrease from ~7.4 to ~6.4 during UV irradiation (Supplementary Fig. 31). This agrees with the apparent p$K_a$ of ~6.5 for PEO$_{45}$-$b$-PPA$_{26}$ (Supplementary Fig. 25), the chemical structure of which is the same as decaged PEO$_{45}$-$b$-PNCMA$_{30}$ (Supplementary Fig. 5). On the other hand, for PEO$_{45}$-$b$-P(NCMA$_{0.55}$-$co$-DPA$_{0.45}$)$_{29}$ polymersome dispersion, UV irradiation only resulted in slight pH decrease from ~7.4 to ~7.2 (Supplementary Fig. 31), further confirming internal proton transfer and ion-pair formation, i.e., light-triggered polymersome-to-PICsome transition.

For decaged PEO$_{45}$-$b$-P(NCMA$_{0.55}$-$co$-DPA$_{0.45}$)$_{29}$ directly generated via UV irradiation in DMSO solution, its self-assembly was actuated by using slow water addition method. As shown in Supplementary Fig. 32, we could only observe the formation of PIC micelles of non-uniform size distribution. This is in distinct contrast to UV-triggered PICsome formation (Fig. 3b), revealing the importance of pre-organization within precursor polymersome bilayers before triggered PICsome formation.

**PICsome nanostructures stabilized by multiple interactions.** The microstructural stability of both polymersomes and PICsomes is a prerequisite towards their potential applications in complex biological milieu[1–5,29–31]. We then examined the stability of polymersomes and resultant PICsomes towards pH, temperature and high ionic strength. As shown in Fig. 6a, original polymersomes (without UV irradiation) exhibited microstructural integrity over the pH range of 5–12; whereas below pH 5, vesicle disintegration occurred due to protonation of tertiary amine comonomers within bilayers (Supplementary Fig. 26, left column).

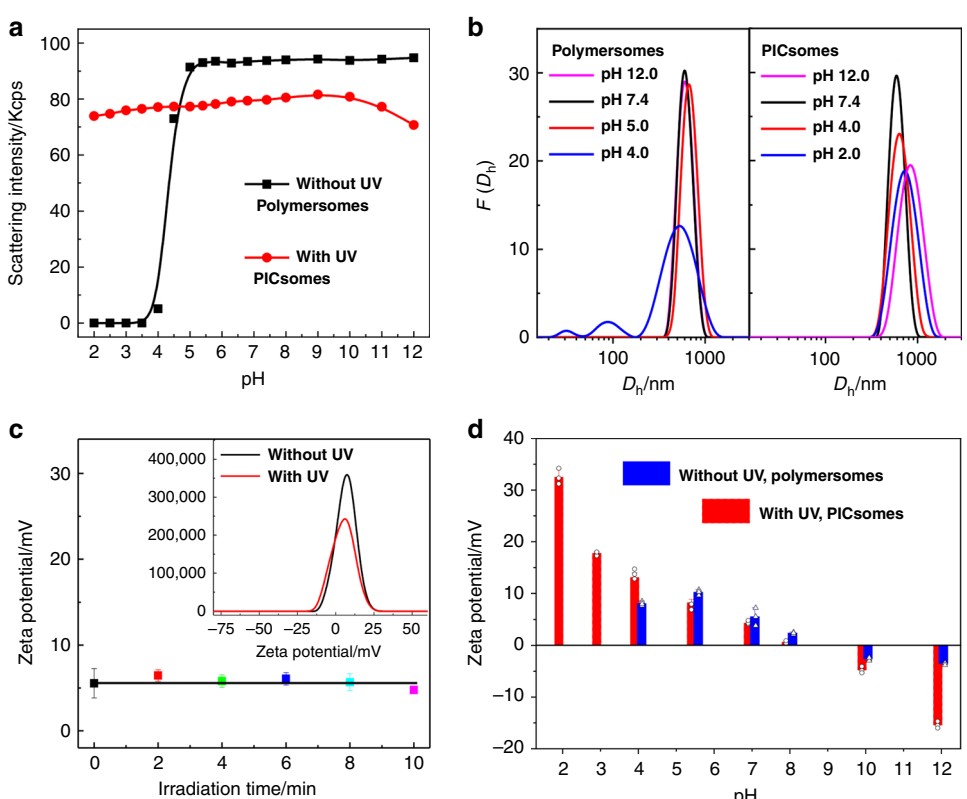

**Fig. 6 Characterization of polymersomes and corresponding PICsomes.** pH-dependent scattered light intensities **a** and intensity-average $D_h$ distributions **b** recorded for aqueous dispersions of PEO$_{45}$-$b$-P(NCMA$_{0.55}$-$co$-DPA$_{0.45}$)$_{29}$ polymersomes and PICsomes in the pH range of 2–12. **c** Irradiation duration-dependent evolution of zeta potentials during polymersome-to-PICsome transition in neutral aqueous media. **d** Variation of zeta potentials for aqueous dispersions of polymersomes and PICsomes in the pH range of 2–12. All data were obtained at a polymer concentration of 0.1 g/L with [NCMA] ~ 0.12 mM and [DPA] ~ 0.10 mM in Britton–Robinson buffer (pH 2–12, 12 mM; 25 °C). The error bars indicate standard deviation ($n = 3$).

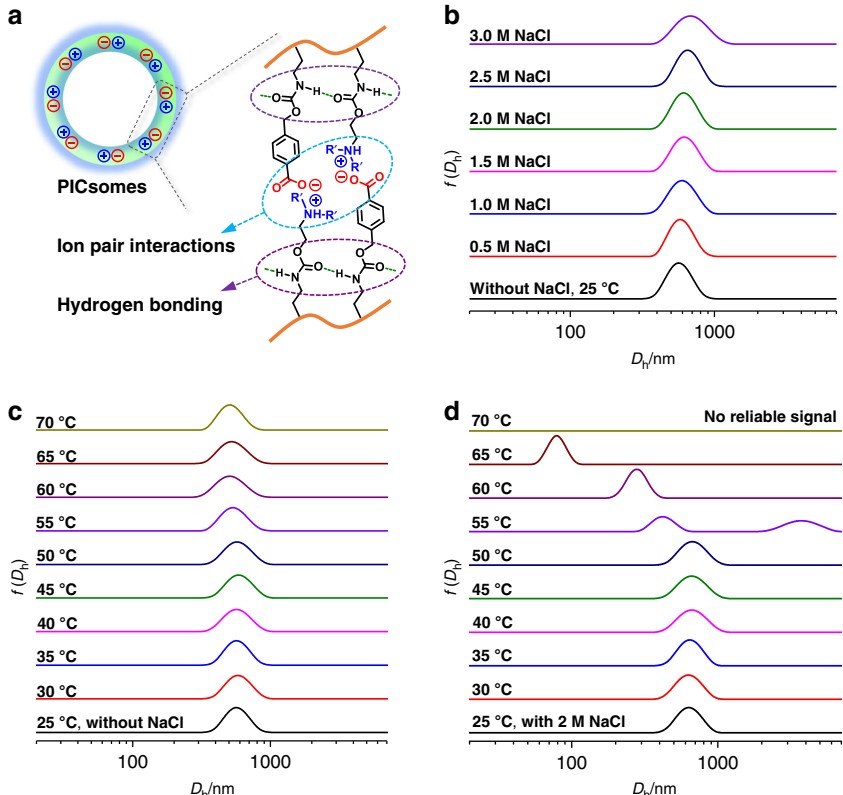

**Fig. 7 PICsome nanostructures stabilized by multiple and cooperative noncovalent interactions. a** Microstructural stability of PICsomes contributing from multivalent ion-pair, hydrogen bonding, and π–π interactions at high local concentration due to the pre-organized nature of vesicle bilayers. **b–d** Intensity-average $D_h$ distributions recorded for the aqueous dispersion of PICsomes (fabricated from PEO$_{45}$-b-P(NCMA$_{0.55}$-co-DPA$_{0.45}$)$_{29}$ polymersomes via UV irradiation for 10 min) at varying temperatures, ionic strengths, and combinations of them. All data were obtained at a polymer concentration of 0.1 g/L with [NCMA] ~ 0.12 mM and [DPA] ~ 0.10 mM in neutral aqueous media.

As for PICsomes obtained via UV irradiation, its structural integrity was maintained over the pH range of 2–12, as confirmed by pH-dependent $D_h$ distributions (Fig. 6b). Meanwhile, TEM observations confirmed that original polymersomes remained to be stable in neutral and alkali media, but disassembled into irregular aggregates under acidic media (pH < 4.0) (Fig. 3c, e). However, after UV irradiation, the resultant PICsomes exhibited highly microstructural stability in the pH range of 2–12 (Fig. 3d, f).

Microstructural stability of PICsomes was further corroborated by zeta potential measurements during the process of polymersome-to-PICsome transition and against varying pH conditions. In neutral aqueous media, zeta potential values remained almost constant to be ca. +6 mV during UV irradiation (Fig. 6c). In the pH range of 7–12, zeta potentials of original polymersomes are in the range of ±5 mV. It slightly increased to ~+8 mV at pH 4, and no detectable data could be obtained when further decreasing pH to ~2–3 due to microstructural disassembly. As for PICsomes, the pH-dependent variation of zeta potentials in the pH range of 4–10 is similar to that of polymersomes. However, in the more acidic and alkaline media (pH 2 and 12), zeta potential values exhibited significant changes (+32 mV at pH 2 and −15 mV at pH 12), which demonstrated the structural integrity of PICsomes. The results are in good agreement with TEM and DLS characterization results (Figs. 3 and 6a).

The stability of PICsomes under extreme pH conditions is quite unexpected, considering that electrostatic interactions of ion-pairs will be considerably weakened due to carboxyl protonation and amine deprotonation in strongly acidic and alkaline media, respectively (Supplementary Figs. 26–29). We tentatively ascribe the observed pH stability to the following two possible reasons[48,50,52,56]: (i) HB interactions between carboxyl moieties in the protonated state, together with cooperative HB and π–π interactions between benzyl carbamate side linkages, and HB interactions between carboxyl and carbamate moieties, could explain the stability at pH 2; (ii) concerning the stability under alkaline media, HB and π–π interactions between benzyl carbamate side linkages should be mainly responsible, in addition to the hydrophobic nature of non-protonated amine comonomer units (Fig. 7a).

Encouraged by the above pH stability of PICsomes, we further investigated the stability of PICsomes towards high ionic strengths and elevated temperatures. After obtaining PICsomes in situ via UV irradiation, different amount of NaCl salt was added, with the final concentration in range of 0–3.0 M. DLS results revealed that as-prepared PICsomes are very stable up to an NaCl concentration of 3.0 M, and $D_h$ distributions remain almost the same with the variation of NaCl concentrations (Fig. 7b). In addition, PICsomes in neutral aqueous media are also stable towards elevated temperatures (25–70 °C); again, $D_h$ distributions exhibit negligible changes in the temperature range investigated (Fig. 7c).

Next, we challenge PICsomes with a combination of salt and elevated temperatures. In the presence of 2.0 M NaCl, PICsomes are stable up to ~50 °C. Further heating to even higher temperatures leads to the evolution of $D_h$ distributions towards lower size ranges. At 70 °C, PICsomes disassemble into unimers and no reliable DLS signals could be detected (Fig. 7d and Supplementary Fig. 33). This is quite reasonable considering that

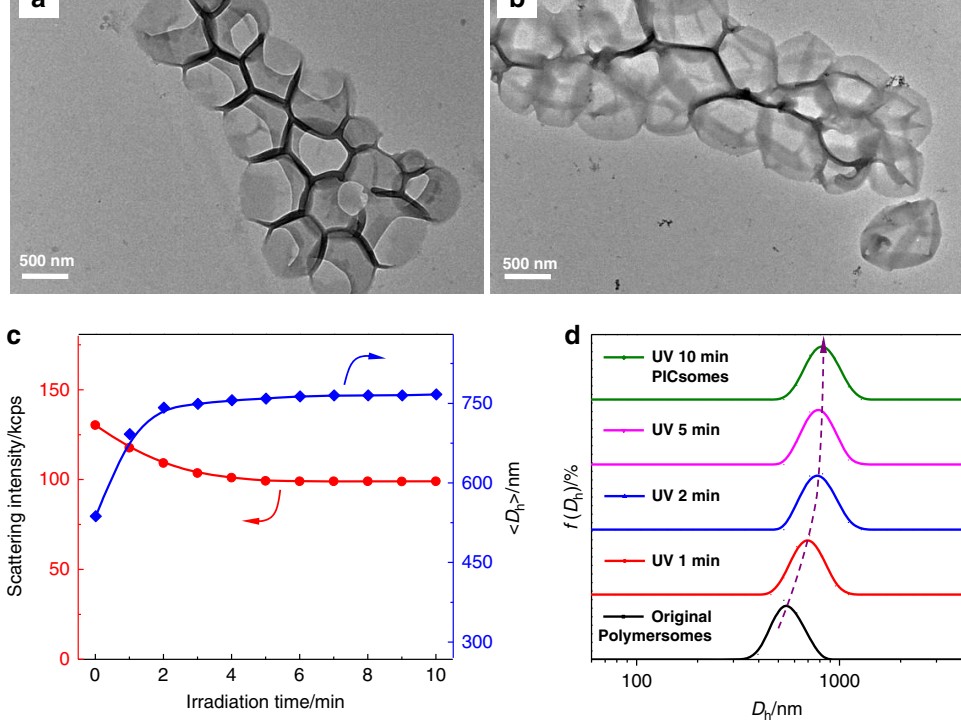

**Fig. 8 Light-triggered microstructural evolution of triblock copolymer vesicles.** TEM images of $PEO_{45}$-$b$-$PNCMA_{17}$-$b$-$PDPA_{21}$ polymersomes **a** and corresponding PICsomes **b**. **c** Irradiation duration-dependent evolution of scattered light intensities and $\langle D_h \rangle$ recorded during polymersome-to-PICsome transition for $PEO_{45}$-$b$-$PNCMA_{17}$-$b$-$PDPA_{21}$ vesicles in neutral aqueous media. **d** Evolution of intensity-average $D_h$ distributions upon UV irradiation. All scale bars are 500 nm. All data were obtained at a polymer concentration of 0.1 g/L in neutral aqueous media.

ionic interactions and HB interactions will be largely suppressed by high salt concentrations and elevated temperatures. The disintegration of PICsomes at 70 °C in the presence of 2.0 M NaCl also confirmed that they are stabilized by cooperative noncovalent interactions, instead of chemical crosslinking (Fig. 1). Note that this is also in agreement with [1]H NMR characterization data recorded in DMSO-$d_6$ for lyophilized polymersome dispersion after being subjected to UV irradiation (Supplementary Fig. 19)[48,50]. It is interesting to note that original polymersomes (without UV irradiation) are very stable towards both high temperature (70 °C) and high ionic strength (2.0 M), and a combination of them (Supplementary Fig. 34), which should also be ascribed to cooperative carbamate-relevant HB interactions and the hydrophobic nature of vesicle bilayers (i.e., incompatible with NaCl salt).

**Effects of block sequences on permselectivity regulation.** As demonstrated above, robust PICsome nanostructures could be in situ fabricated from $PEO_{45}$-$b$-$P(NCMA_{0.55}$-$co$-$DPA_{0.45})_{29}$ polymersomes in aqueous media via UV-triggered carboxyl decaging. The bilayer forming block is a random copolymer of DPA and NCMA. We speculated that if DPA and NCMA comonomers are arranged in the block instead of random sequence, UV decaging of NCMA will afford triblock poly-ampholytes consisting of two oppositely charged blocks[43,44], which is more comparable to conventional PICs[29–31,45–47]. $PEO_{45}$-$b$-$PNCMA_{17}$-$b$-$PDPA_{21}$ with chemical compositions comparable to that of $PEO_{45}$-$b$-$P(NCMA_{0.55}$-$co$-$DPA_{0.45})_{29}$ BCP was then synthesized (Fig. 2).

According to similar procedures used for the diblock copolymer, $PEO_{45}$-$b$-$PNCMA_{17}$-$b$-$PDPA_{21}$ triblock copolymer

also self-assembled into polymersomes with quite uniform size distribution, as determined by TEM (Figs. 1 and 8a). Accordingly, UV irradiation of the triblock polymersome dispersion also generated carboxyl and ion-pairs within bilayers, leading to polymersome-to-PICsome transition. As shown in Fig. 8b, the vesicular nanostructure was well maintained, and the overall dimension of resultant PICsomes is larger than that of original polymersomes. DLS measurements indicated that the $\langle D_h \rangle$ of vesicles exhibited a rapid increase within the initial ~3–4 min UV light irradiation; the increase of $\langle D_h \rangle$ from 540 to 770 nm during polymersome-to-PICsomes transition (Fig. 8c), which is quite prominent compared to that of $PEO_{45}$-$b$-$P(NCMA_{0.55}$-$co$-$DPA_{0.45})_{29}$ vesicles (Fig. 4a; from 550 to 570 nm upon UV irradiation). It is worthy of noting that during the whole transition process, the vesicle size distribution remained to be quite uniform ($\mu_2/\Gamma^2 \sim 0.10$; Fig. 8d). To verify the importance of pre-organization within bilayers of precursor polymersomes before triggered PICsome formation, $PEO_{45}$-$b$-$PNCMA_{17}$-$b$-$PDPA_{21}$ was subjected to direct decaging upon UV irradiation in DMSO, followed by self-assembly in aqueous media. We could only observe the formation of micellar nanoparticles instead of polymeric vesicles (Supplementary Fig. 35).

Drastically different extents of swelling during UV-actuated polymersome-to-PICsome transition for $PEO_{45}$-$b$-$P(NCMA_{0.55}$-$co$-$DPA_{0.45})_{29}$ diblock and $PEO_{45}$-$b$-$PNCMA_{17}$-$b$-$PDPA_{21}$ tri-block copolymers reflected the effects of comonomer sequences upon formation of ion-pairs (Figs. 1 and 2). For diblock vesicles, newly generated carboxyl functionalities are in close proximity with tertiary amine moieties due to the random copolymerization nature. Thus, the local concentration of ion-pair interactions will be very high. In contrast, for triblock vesicles, initially generated carboxyl functionalities are more apart from amine moieties,

leading to frustrated formation of ion-pairs and bilayer swelling due to partial ionization and protonation of carboxyl/amine residues (Fig. 8c). Only at later stages of UV irradiation, PIC formation between oppositely charged blocks (i.e., decaged NCMA block and DPA block) will occur, prohibiting further bilayer swelling. This explains that vesicle sizes remain almost constant after ~4 min UV irradiation.

UV-triggered polymersome-to-PICsome transition should be accompanied with the transformation of bilayer polarity from being hydrophobic to hydrophilic due to the loss of hydrophobic 2-nitrobenzyl ester residues and generation of ion-pairs. We further examined this issue by using polymersomes covalently conjugated with a microenvironmental polarity-sensitive fluorescent probe, i.e., Nile red. As shown in Supplementary Fig. 36, for polymersomes co-assembled from $PEO_{45}$-$b$-P($NCMA_{0.55}$-$co$-$DPA_{0.45}$)$_{29}$ and $PEO_{45}$-$b$-P($NCMA_{0.55}$-$co$-$DPA_{0.45}$)$_{29}$-$Nile$ $red$ (8:2, molar ratio) in aqueous media, the initial Nile red emission at ~610 nm is quite strong, indicating the hydrophobic nature of polymersome bilayers. Upon UV irradiation, considerable decrease of emission intensities is clearly evident, accompanied with slight red shift of emission maxima. This confirms the generation of hydrophilic PICsome bilayers upon UV irradiation (Fig. 1). For diblock and triblock copolymer vesicle discussed above, different extents of swelling during polymersome-to-PICsome transition also hint varying mesh sizes and bilayer permeability for resultant PICsomes.

To probe changes in bilayer permeability, we loaded several types of water-soluble anticancer drugs in the aqueous interior of polymersomes and examined release profiles upon UV light-actuated polymersome-to-PICsome transition (Fig. 9a). For gemcitabine hydrochloride (299.7 Da) loaded within polymersomes of $PEO_{45}$-$b$-P($NCMA_{0.55}$-$co$-$DPA_{0.45}$)$_{29}$ without UV irradiation, ~25% cumulative release was observed over 32 h (Fig. 9b). Upon UV irradiation for 2 and 5 min, sustained release of gemcitabine hydrochloride was achieved, with cumulative release extents being ~85% and ~97% after 32 h, respectively. However, doxorubicin hydrochloride (Dox·HCl) with larger molar mass (580.0 Da), only <5% drug could be released over 32 h from original polymersomes, and PICsomes upon UV irradiation (Fig. 9c). As both gemcitabine hydrochloride (299.7 Da) and Dox·HCl bear positive charges, the above discrepancy of release profiles should be ascribed to different molecular sizes, suggesting the excellent permselectivity of resultant PICsomes. This is in stark contrast to conventional PICsomes, from which the release of dextran with molar mass up to ~10 kDa could still be released[30].

We further checked permeabilities of neutral and negatively charged drugs through PICsome bilayers. As shown in Supplementary Fig. 37, neutral 2′-deoxy-5-fluorouridine (5-Fu; 246.2 Da) and negatively charged coumarin-343 (model drug; 285.3 Da) could be on-demand released from vesicles of $PEO_{45}$-$b$-P($NCMA_{0.55}$-$co$-$DPA_{0.45}$)$_{29}$ via triggered polymersome-to-PICsome transition. On the other hand, the release of negative charged calcein with higher molar mass (622.6 Da) from both polymersomes and PICsomes are completely prohibited. The above results indicated that the polymersome-to-PICsome transition is accompanied with permeability switching of vesicle bilayers and molecular size-selective release of encapsulated drugs could be successfully achieved. This could be ascribed to the presence of paired ionic interactions at high local concentration within PICsome bilayers, providing accurate mesh size control by physical crosslinking through a combination of electrostatic ion-pair interactions and side chain HB interactions.

For polymersomes of $PEO_{45}$-$b$-PNCMA$_{17}$-$b$-PDPA$_{21}$ triblock copolymer, the hydrophobic-to-hydrophilic transition of bilayers also occurs during light-triggered polymersome-to-PICsome transition, as confirmed by fluorescence measurements of physically encapsulated Nile red probe (Supplementary Fig. 38). In contrast to $PEO_{45}$-$b$-P($NCMA_{0.55}$-$co$-$DPA_{0.45}$)$_{29}$ vesicles, PICsomes of the triblock copolymer exhibited on-demand sustained release for gemcitabine hydrochloride, Dox·HCl, and calcein (Fig. 9d–f and Supplementary Fig. 39); note that PICsomes of the former exhibited almost no release of Dox·HCl and calcein (Fig. 9d and Supplementary Fig. 37). It is also intriguing to note that release rates of gemcitabine hydrochloride, Dox·HCl, and calcein from triblock PICsomes (5 min UV irradiation) decreased in the order of increasing molar mass and number of charges (~100% cumulative release of gemcitabine hydrochloride over 16 h; ~84% Dox·HCl release over 32 h, and ~49% calcein release over 32 h, respectively). The dramatically enhanced permeability of PICsomes of the triblock copolymer compared to that of diblock copolymer could be safely ascribed to the significant swelling (from 540 to 770 nm; ~43% increase in $\langle D_h \rangle$) during light-triggered polymersome-to-PICsome transition; whereas diblock vesicles only exhibited an $\langle D_h \rangle$ increase ~3.6% (from 550 to 570 nm). The block sequence-dependent permselectivity of in situ fabricated PICsomes from precursor polymersomes augurs well for their practical applications as both drug nanocarriers and nanoreactors. In addition, UV irradiation duration could serve as another dimension to modulate bilayer permeability and selectivity, with longer UV irradiation affording accelerated release of encapsulated functional agents (Fig. 1). To the best of our knowledge, the permselectivity modulation by both block copolymer sequences and magnitude of external stimuli has not been achieved before.

**Reduction-triggered polymersome-to-PICsome transition.** The previous sections established that light-triggered polymersome-to-PICsome transition affords ultrastable vesicles with excellent bilayer permselectivity towards a series of drug molecules of varying molar mass and number of charges. We further generalized the design by actuating polymersome-to-PICsome transition using reductive milieu trigger (Fig. 1). Note that the redox gradient across cell membranes is universal for all natural organisms[14,57,58]. For $PEO_{45}$-$b$-P($DCMA_{0.45}$-$co$-$DPA_{0.55}$)$_{33}$ diblock copolymer containing disulfide-caged carboxyl comonomers (DCMA; Fig. 2), its self-assembly in aqueous media again afforded polymersomes of ~600 nm in diameter, with quite uniform size distribution (Fig. 10a).

Reduction-triggered evolution of vesicular microstructures was then explored by TEM and DLS measurements (Fig. 10b–d). Upon GSH addition, disulfide cleavage is accompanied with spontaneous 1,6-rearrangement of benzyl moieties[14,57–59], generating carboxyl functionalities and leading to ion-pair formation (Fig. 1). The reduction-triggered decaging process was further examined with $^1$H NMR, exhibiting a decaging extent of >99% upon treating with 10 mM GSH for 24 h (Supplementary Fig. 40). Although we could observe the apparent decrease of scattered light intensities, $\langle D_h \rangle$ distributions remained almost unchanged upon treating with GSH; the vesicular integrity was further confirmed by TEM observations (Fig. 10b). Thus, although the reductive trigger and light trigger act on different time scales (hours vs. minutes), both could actuate polymersome-to-PICsome transition, with the formation of stable PICsomes with hydrophilic bilayers.

Finally, we examined the permselectivity of in situ-fabricated PICsomes of $PEO_{45}$-$b$-P($DCMA_{0.45}$-$co$-$DPA_{0.55}$) upon actuating with reductive trigger (i.e., GSH). As shown in Fig. 10e, encapsulated model anticancer drug (5-Fu) exhibited retarded release from both original polymersomes and those treated with ~2 μM GSH (comparable to extracellular and blood circulation

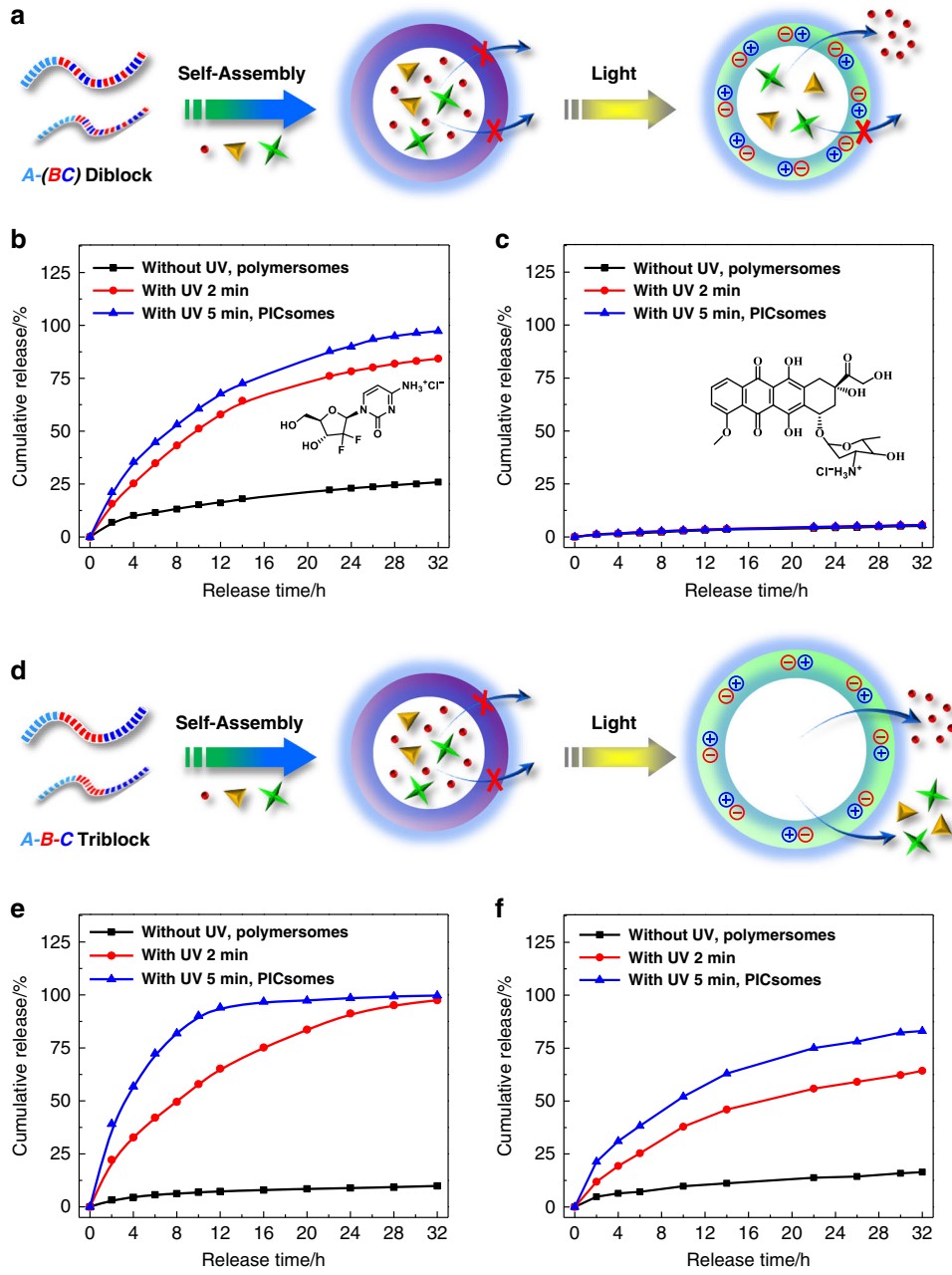

**Fig. 9 Light-regulated polymersome-to-PICsome transition for molecular size-selective drug release. a, d** Schematics of drug-loaded vesicles of PEO$_{45}$-*b*-P(NCMA$_{0.55}$-*co*-DPA$_{0.45}$)$_{29}$ diblock copolymer **a** and PEO$_{45}$-*b*-PNCMA$_{17}$-*b*-PDPA$_{21}$ triblock copolymer **d**, and triggered release from corresponding PICsomes. **b, c, e, f** Release profiles of encapsulated **b, e** gemcitabine hydrochloride and **c, f** Dox·HCl from the aqueous lumen of PEO$_{45}$-*b*-P(NCMA$_{0.55}$-*co*-DPA$_{0.45}$)$_{29}$ diblock copolymer **b, c** and PEO$_{45}$-*b*-PNCMA$_{17}$-*b*-PDPA$_{21}$ triblock copolymer **e, f** vesicles before and after UV irradiation. All data were obtained at a polymer concentration of 0.1 g/L in PB buffer (pH 7.4, 10 mM, 37 °C).

milieu), with cumulative release extents being ~14% and ~19% over 32 h, respectively. On the other hand, upon treating with 5 and 10 mM GSH (comparable to cytosolic milieu), up to ~73% and ~94% 5-Fu release was achieved over 32 h incubation duration. However, in the case of loaded Dox·HCl, prohibited release (<8% cumulative release over 42 h) was observed for both original polymersomes and GSH-treated ones (Fig. 10f). This feature is quite similar to that of PEO$_{45}$-*b*-P(NCMA$_{0.55}$-*co*-DPA$_{0.45}$)$_{29}$ vesicles upon light-triggered polymersome-to-PICsome transition. Thus, irrespective of the types of external triggers (UV light or GSH), the permeability switching and bilayer permselectivity is mainly determined by the sequence of bilayer-formation block (e.g., random vs. block).

## Discussion

Polymersomes and PICsomes, with distinct bilayer compositions and polarity, represent two main categories of vesicular nanostructures self-assembled from BCPs. However, polymersomes possess retarded bilayer permeability, whereas PICsomes with semipermeable membrane are unstable towards physiologically relevant salt and temperature. We proposed a general strategy to solve this challenge by external stimuli-triggered polymersome-to-PICsome transition, starting from a single component functionalized BCP precursor. In situ-fabricated PICsomes via UV light trigger are ultrastable towards extreme pH (2–12), elevated temperature (up to 70 °C), and high ionic strength (3 M NaCl), due to cooperative ion-pair interactions at high local concentration and

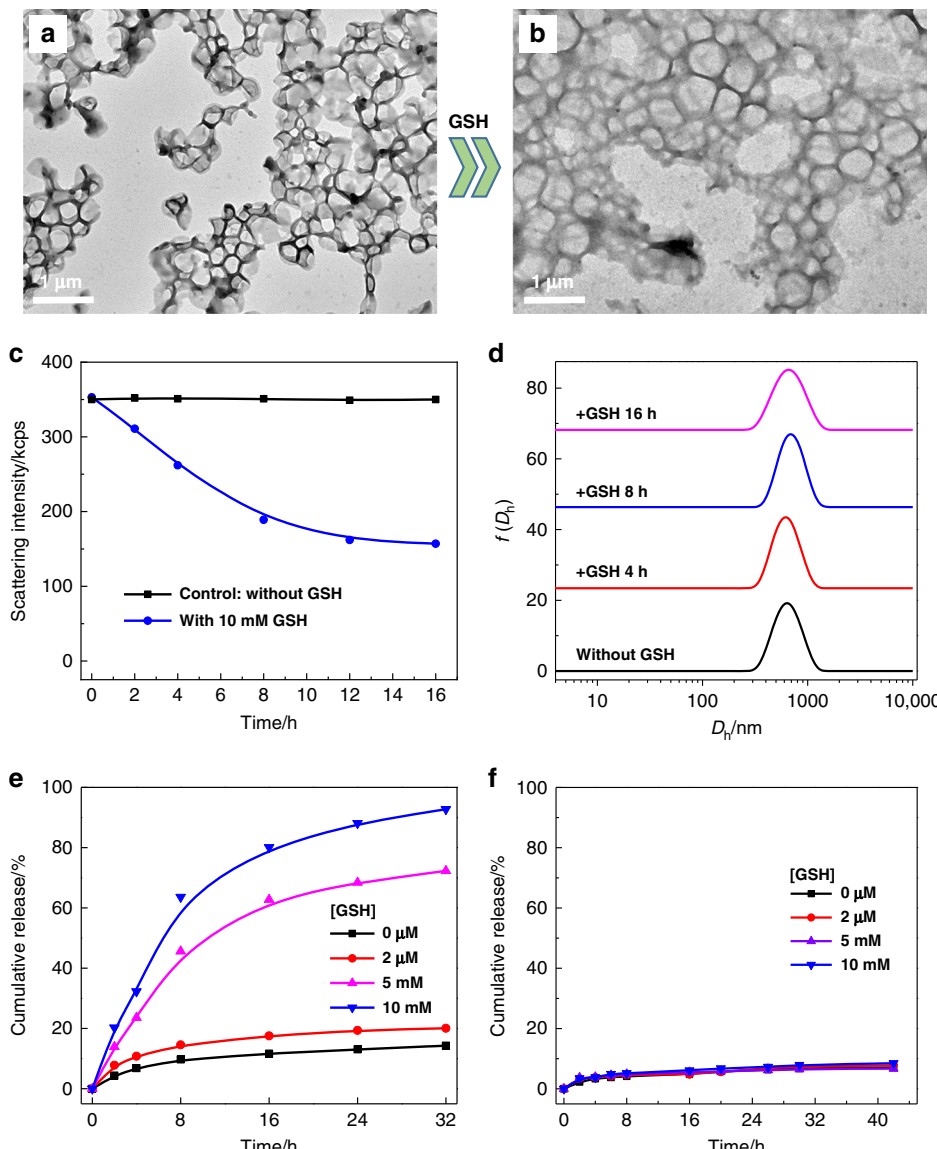

**Fig. 10 Reduction-triggered polymersome-to-PICsome transition and permselectivity regulation. a, b** TEM images recorded for reductive milieu-responsive PEO$_{45}$-*b*-P(DCMA$_{0.45}$-*co*-PDPA$_{0.55}$)$_{33}$ vesicles **a** before and **b** after treating with 10 mM GSH in neutral aqueous media. **c, d** Incubation time-dependent evolution of **c** scattered light intensities and **d** intensity-average $D_h$ distributions upon treating with 10 mM GSH. **e, f** Release profiles of **e** 5-Fu and **f** Dox·HCl from aqueous interiors of vesicles before and after treating with GSH at varying concentrations. All scale bars represent 1 μm. All data were obtained at a polymer concentration of 0.1 g/L in PB buffer (pH 7.4, 10 mM, 37 °C).

synergistic carbamate-relevant HB interactions within vesicle bilayers. The polymersome-to-PICsome transition is accompanied with prominent hydrophobic-to-hydrophilic permeability switching. As compared to that of conventional PICsomes fabricated from two oppositely charged block polyelectrolytes, the excellent permselectivity demonstrated by in situ-fabricated PICsome bilayers represents another important feature, which could be further regulated by comonomer sequences of the bilayer-forming block (random vs. block) and the magnitude of external stimuli. The proposed strategy of triggered polymersome-to-PICsome transition combines advantages of two main types of BCP vesicles and solves the stability issue of PICsomes without recourse to chemical crosslinking. In addition, reductive milieu could also be utilized to trigger polymersome-to-PICsome transition, auguring well for the generality of the proposed in situ transformation strategy.

## Methods

**Sample synthesis**. Synthetic routes employed for the preparation of 2-nitrobenzyl ester-photocaged carboxyl monomer (NCMA), DCMA, and two types of tertiary amine-containing monomer with carbamate linkages, DPA and DEA, are shown in Supplementary Fig. 1. Schematics of the synthesis of UV light-responsive PEO$_{45}$-*b*-P(NCMA$_x$-*co*-DPA$_{1-x}$)$_n$ diblock copolymers and PEO$_{45}$-*b*-PNCMA$_m$-*b*-PDPA$_n$ triblock copolymers, and disulfide-caged PEO$_{45}$-*b*-P(DCMA$_x$-*co*-DPA$_{1-x}$)$_n$ diblock copolymer are shown in Supplementary Figs. 2 and 3, respectively. Schematics of the synthesis of dye-functionalized amphiphilic diblock copolymers, PEO$_{45}$-*b*-P(NCMA$_x$-*co*-DPA$_{1-x}$)$_n$-*Nile red* and PEO$_{45}$-*b*-P(NCMA$_x$-*co*-DPA$_{1-x}$)$_n$-*naphthali-mide*, and two types of control diblock copolymers without tertiary amine moieties, PEO$_{45}$-*b*-PNCMA$_{30}$ and PEO$_{45}$-*b*-PPA$_{26}$, are shown in Supplementary Figs. 4 and 5, respectively. Detailed procedures of sample synthesis and structural character-ization data are described in the Supplementary Information.

**Self-assembly of amphiphilic BCPs**. In a typical self-assembling procedure, 2 mg amphiphilic block copolymer was dissolved in 1 mL acetone, stirred and thermo-stated at 25 °C in a water bath. Next, 9 mL deionized water was slowly added over 9 h via a springe pump. The organic solvent was then removed by dialysis (MWCO

3.5 kDa) against deionized water for 8 h and the external dialysate was replaced with fresh deionized water at an ~2 h interval.

**Fabrication of dye-labeled vesicles**. Dye-labeled vesicles were fabricated via co-assembly of label-free amphiphilic diblock copolymers with dye-functionalized amphiphilic BCPs including P(NCMA$_{0.55}$-co-DPA$_{0.45}$)$_{29}$-*naphthalimide* and PEO$_{45}$-b-P(NCMA$_{0.55}$-co-DPA$_{0.45}$)$_{29}$-*Nile red*. In a typical procedure employed for the fabrication of vesicles labeled with pH-sensitive naphthalimide-based probes, PEO$_{45}$-b-P(NCMA$_{0.55}$-co-DPA$_{0.45}$)$_{29}$ (1.6 mg) and P(NCMA$_{0.55}$-co-DPA$_{0.45}$)$_{29}$-*naphthalimide* (0.4 mg) were dissolved in 1 mL acetone, stirred and thermostated at 25 °C in a water bath. Then, 9 mL water was slowly added within 9 h via a springe pump. The organic solvent was then removed by dialysis (MWCO 3.5 kDa) against deionized water for 8 h and the external dialysate was replaced with fresh deionized water at an ~2 h interval.

**Fabrication of drug/model drug-encapsulated vesicles**. For the physical encapsulation of hydrophobic Nile red into hydrophobic bilayers of self-assembled polymersomes, the amphiphilic block copolymer and Nile red were dissolved in acetone at final concentrations of 2.0 and 0.01 g/L, respectively. The solution mixture was then subjected to similar self-assembling procedures described above.

For the encapsulation of hydrophilic drug and model drug molecules (e.g., anticancer drug 2′-deoxy-5-fluorouridin, 5-Fu) into the hydrophilic lumen of self-assembled polymersomes, 2.0 mg amphiphilic diblock copolymer was dissolved in 1 mL acetone, stirred and maintained at 25 °C in a water bath. Then, 5-Fu (16 mg, 125 μmol) dissolved in 9 mL water was slowly added within 9 h. The organic solvent was then removed by dialysis (MWCO 3.5 kDa) against deionized water for 8 h and the external dialysate was replaced with fresh deionized water at ~2 h interval. According to similar procedures, other water-soluble anticancer drugs and model drugs including gemcitabine hydrochloride, doxorubicin hydrochloride (Dox·HCl), coumarin 343, and calcein were also encapsulated into the aqueous lumen of polymersomes, and vesicular self-assembly was actuated by slowing adding aqueous solution of corresponding drugs at the same molar concentration. The photostability assay results of drugs and model drugs against UV light irradiation are shown in Supplementary Fig. 41. Drug loading efficiency, loading content, and loaded drug concentration were quantified by fluorescence (Dox·HCl, coumarin 343, and calcein) and UV–Vis absorbance (5-Fu and gemcitabine hydrochloride), respectively. Relevant results are summarized in Supplementary Table 2.

## Data availability

The data in this work are available in the manuscript or Supplementary Information, or available from the corresponding author upon request. The source data underlying Figs. 4a, b, 5b, c, 6a–d, 7b–d, 8c, d, 9b, c, e, f, 10c–f and Supplementary Figs. 16, 18a, b, 19c, 20a, b, 21, 22c, d, 23, 24, 25, 27, 28b, c, 29b, c, 30a, b, 31c, 32c, 33, 34, 35c, 36b, c, 37a, b, 38, 39, 40c, 41a–e are provided as a Source Data file.

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

## Acknowledgements

The financial support from National Natural Science Foundation of China (NNSFC) Project (51690150, 51690154, 21674103, 21905130, 51773190, and U19A2094) and International S&T Cooperation Program of China (ISTCP) of MOST (2016YFE0129700) is gratefully acknowledged.

## Author contributions

X.R.W. and S.Y.L. conceived the project and designed the experiments. S.Y.L. supervised and supported the project. X.R.W. and C.Z.Y. developed the materials and performed the characterization. X.R.W., G.Y.Z., and S.Y.L. analyzed the data. X.R.W. and S.Y.L. wrote the paper.

## Competing interests

The authors declare no competing interests.
