## [Peer Review File · Nature Communications]

Reviewers' Comments:

Reviewer #1:

Remarks to the Author:

This manuscript describes a new way of making polyion complex vesicles (PICsomes) and an extensive characterization of their stability and release properties. The premise of the manuscript is that assembly of block copolymersomes stabilized by hydrophobic interactions including caged charged groups that are then converted to charged groups using UV irradiation creates stable yet permeable vesicles. The trigger to the permeability change is not applied in situ; it is a preparation step.

The approach is definitely very interesting and extensive investigations were performed, the results are conclusive and partly impressive. Methodologically the work is very sound. In my view, the material could be well suited for Nature Communications.

The main issue that I find important to resolve is the characterization of the product and the comparison of other ways to achieve the identical type of PICsomes using direct self-assembly rather than the current two-step approach. I find that important for the understanding and for drawing the right conclusions from the work.

Currently, the effect of UV irradiation on secondary properties such as size, zeta potential and response to potentiometric titration is followed. The end product should be characterized also by chemical methods to establish the conversion yield and whether other effects on the polymer composition and structure is found. Most likely only average properties can be found, but a detailed NMR and mass spec characterization would be advised.

It is not clear to me why the process should lead to higher stability, the latter being a thermodynamic property. A question, therefore, is if it is a combination of remaining hydrophobic interactions and the polyion interactions that create the interesting balance of high stability and high permeability for small Mw compounds that is observed. All comparisons are performed in relation to the polymersomes, but a comparison to equivalent PICsomes formed in a different way and claimed have too low stability is largely missing.

Minor comments to the possible impact of the concept is that the vesicles that are investigated are more than half a micron big. That is generally considered too large to be efficient for most drug delivery applications. I am sure there are possibilities to resize the vesicles that that could be investigated to remedy this. Ideally that is done, however, it would be possible to just comment on this and how it would be addressed. Additionally, several compounds are encapsulated and their long-term release profiles are investigated. Here, the release rate is more interesting than the release fraction, which in many cases were quite similar between the block copolymersomes and the PICsomes. The authors also present that data only in relative release. A description of the absolute amounts that could be incorporated into the vesicles should be included for relevance to drug delivery applications.

The TEM images indicate a significant fraction of smaller objects that are not captured by the DLS size distributions that are very narrow. Comment?

In which way does Figure 7e show "microstructural destruction"?

Reviewer #2:

Remarks to the Author:

In this submission, Liu and coauthors describe a general method to convert a polymersome to PICsome in situ using stimuli as a handle. This transformation is accompanied with permeability switching from impermeable hydrophobic membrane to semipermeable molecular size-selective membrane. The permeability of the PICsome can be finely tuned by changing the composition of block copolymer and duration of stimuli treatment. The molecular size-selective permeability was further utilized to selectively release cancer drugs with small size difference. The resulted PICsome shows high stability in broad pH range, strong ionic strength, and high temperature, which is not common in classic PICsomes. This high stability is attributed to the cooperative effect of hydrogen bonding and ionic pairing. Overall, it is an interesting concept to achieve the semipermeable

nanomembrane. However, following concerns need to be addressed.

1. The polymersome to PICsome transitions all rely on the assumption that decaging of carboxylate protonates the tertiary amine and consequently forms ionic pairs. However, the characterization on the decaging is not solid enough to prove the transformation. The decaging of 2-nitrobenzyl ester was only followed by monitoring the generation of nitroso benzaldehyde in absorption spectrum. But, there is almost no increase in absorbance at 365 nm after UV irradiation (Supplementary Fig. 17). Photolysis of 2-nitrobenzyl ester is well-known to be non-efficient in literatures, which is also supported by the NMR experiment in Supplementary Fig 18. NMR of UV-treated polymersome should be taken in the organic NMR solvent to quantify the conversion of 2-nitrobenzyl to nitroso benzyl aldehyde. The percentage of remaining caged carboxylate can significantly change the pKa resulted polymers. Without clear quantification of the functional group transformation, later discussion on PICsome properties relying on polymer pKa cannot be trusted. Similarly, the direct evidences need to be given to support decaging of carboxylate by GSH.

2. It doesn't make sense that the PICsome doesn't disassemble in pH2.0 while the corresponding polymersome precursor which is supposed to be more hydrophobic disassembled in pH4.0. In these acidic conditions, the electrostatic interaction should be eliminated in both polymersome and PICsome, and the hydrophobic interaction would dominate the driving force for the assembly. The authors don't give any explanation. Decaged PEO45-b-P(NCMA0.55-co-DPA0.45)29 polymer should be used to form self-assembly in pH2.0. Its morphology should be the same as the morphology of PICsome that is formed in situ in the extreme pH. Also, the irradiation condition and NMR of decaged PEO45-b-P(NCMA0.55-co-DPA0.45)29 polymer should be provided. It is highly likely that irradiation of intrinsically photo-unstable amine containing polymers may induce side reactions that can potentially crosslink the assembled polymer. The PICsome composition and structure need to be rigorously characterized by NMR and GPC.

3. The fluorescence of naphthalimide and Nile red upon light exposure and different pH need to be done as a control experiment.

4. It is not fair to use release rate of encapsulated cargos to compare the permeability of the membrane without knowing how much of cargo is loaded in the assembly. To be conclusive, the encapsulation of different cargos needs to be similar.

Reviewer #3:

Remarks to the Author:

Polymersomes and polyion complex vesicles (PICsomes) are artificial nanostructures mimicking natural membrane structures consisting of aqueous interiors enclosed by hydrophobic bilayers. However, there is a dilemma between polymersomes and PICsomes in that polymersomes possess enhanced stability and compromised membrane permeability, while PICsomes possess semipermeable membranes but low stability towards physiologically relevant ionic strength and temperature. In this manuscript, the authors proposed a strategy to solve this dilemma by stimuli-triggered polymersome-to-PICsome transition with pre-organized block copolymer vesicles and subsequent de-caging reaction. Contrary to conventional PICsomes, the in situ generated ones are highly stable towards extreme pH range (pH 2-12), high ionic strength and elevated temperature, while maintain semipermeable properties. Moreover, the permeability of the PICsomes could be tuned by the sequence structure of the block copolymers, which enabled on-demand triggered release functions.

This is an interesting and highly important study, and provides a novel paradigm for polymeric vesicles design leveraging stimuli-responsive properties. The whole manuscript is in precise logic and the conclusions are fully supported by the data. The manuscript should be accepted for publication in this journal.

The following are some minor points for this manuscript:

1. It is interesting to see the comparison between diblock copolymer and triblock copolymer with

similar components (PEO45-b-P(NCMA0.55-co-DPA0.45)₂₉ vs. PEO45-b-P(NCMA17-b-PDPA21)). The triblock copolymer showed faster drug release upon UV stimulation than diblock copolymer due to the swelling behavior. One possible reason for this difference is the relatively deficient in ion pair between DPA and NCMA in the block copolymer. However, there are more DPA in the triblock copolymer (n=21) than that in the diblock copolymer (n=13). Will extra non-pairing DPA be another reason for the swelling? The NCMA and DPA are almost equal in the studied diblock copolymer (0.55/0.45). Will the NCMA/DPA ratio be another parameter for adjusting the release rate of the diblock polymer vesicles?

2. DOX and some other drugs are sensitive to UV irradiation. Will that affect the release study?
3. The authors used "w/o" for "without", but "w/o" is easy confusing with "oil-in-water".
4. In supplementary Fig.17, what does the blue and red arrow represent?
5. In supplementary Fig.18, the authors used H₂O represent the solvent in the NMR spectrum, there may also be H₂O. Simply use "solvent" may be more accurate.
6. Line147, Supplementary "Fig1b" should be "Fig1c".

RESPONSES TO REFEREES (NCOMMS-19-32638A):
(Reviewer comments in black, our response in blue)

Reviewer #1 (Remarks to the Author):

This manuscript describes a new way of making polyion complex vesicles (PICsomes) and an extensive characterization of their stability and release properties. The premise of the manuscript is that assembly of block copolymerosomes stabilized by hydrophobic interactions including caged charged groups that are then converted to charged groups using UV irradiation creates stable yet permeable vesicles. The trigger to the permeability change is not applied in situ; it is a preparation step. The approach is definitely very interesting and extensive investigations were performed, the results are conclusive and partly impressive.

We appreciate positive comments of Reviewer 1.

Methodologically the work is very sound. In my view, the material could be well suited for Nature Communications.

Q1: The main issue that I find important to resolve is characterization of the product and the comparison of other ways to achieve the identical type of PICsomes using direct self-assembly rather than the the current two-step approach. I find that important for the understanding and for drawing the right conclusions from the work.

Currently, the effect of UV irradiation on secondary properties such as size, zeta potential and response to potentiometric titration is followed. The end product should be characterized also by chemical methods to establish the conversion yield and whether other effects on the polymer composition and structure is found. Most likely only average properties can be found, but a detailed NMR and mass spec characterization would be advised.

Many thanks for insightful comments concerning structural characterization of final nanostructures generated by UV light irradiation. We conducted additional experiments to probe the carboxyl decaging process by ^1H NMR (DMSO- d_6 solvent, lyophilized powder of nanostructure dispersions after being subjected to UV irradiation for 0-10 min). Relevant results are shown in Supplementary Fig. 19, demonstrating a photocleavage extent of >98% upon UV irradiation for 10 min (*page 8, third paragraph*), and the photo-decaging kinetics agrees quite well with the process monitored by UV/Vis absorption spectra (Supplementary Fig. 18). We also attempted to characterize UV-triggered chemical structural changes by matrix-assisted laser desorption/ionization time-of-flight mass spectrometry (MALDI-TOF MS) technique, but no reliable signals could be recorded for both non-irradiated and UV-irradiated polymerosome dispersions upon lyophilization (Supplementary Fig. 20), which might be due to the high molar mass of block copolymer and/or low extent of ionization under MALDI-TOF MS characterization conditions.

Q2: It is not clear to me why the process should lead to higher stability, the latter being a thermodynamic property. A question, therefore, is if it is a combination of remaining hydrophobic interactions and the polyion interactions that create the interesting balance of high stability and high permeability for small Mw compounds that is observed. All comparisons are performed in relation to

the polymersomes, but a comparison to equivalent PICsomes formed in a different way and claimed have too low stability is largely missing.

In the current system, multiple types of noncovalent interactions, including hydrogen bonding, ion-pair, and π - π interactions, cooperatively exist and contribute to the observed high PICsome stability and excellent permselectivity. As shown in Supplementary 19, the photocleavage extent is >98% upon 10 min UV irradiation; thus, hydrophobic interactions associated with 2-nitrobenzyl moieties could be safely excluded. However, hydrophobic interactions associated with polymethacrylate backbones, could still contribute to the observed stability. Corresponding PICsomes are stable towards 3.0 M NaCl or elevated temperature up to 70 °C, but will disassemble when being subjected to a combination of 2.0 M NaCl and >55 °C. These results indicated that the PICsome stability could be mainly ascribed to cooperative ion-pair and hydrogen bonding interactions, the strengths of which are sensitive to high ionic strength and elevated temperature, respectively (Fig. 7 and Supplementary Fig. 33) (pages 14-15).

According to previous literature reports, both polyion complex (PIC) micelles (Alakhov, *et al. Bioconjugate Chem.* **1996**, *6*, 639-643; Kataoka, *et al. Macromolecules* **1995**, *28*, 5294-5299; Eisenberg, *et al. Macromolecules* **1996**, *29*, 6797-6802) and PICsomes (Kataoka, *et al. J. Am. Chem. Soc.* **2006**, *128*, 5988-5989; Kataoka, *et al. J. Am. Chem. Soc.* **2010**, *132*, 1631-1636; Kataoka, *et al. J. Am. Chem. Soc.* **2013**, *135*, 1423-1429) are intrinsically sensitive to high ionic strength, pH, and temperature. In this work, we observed that PICsomes obtained via the two-step strategy, polymersome-to-PICsomes transition, possess excellent stability against high ionic strength, wide range of pH, and temperature. We propose that upon stimuli-triggered decaging (e.g., UV light), vesicular nanostructures are stabilized by newly generated ion-pair interactions at high local concentrations and cooperative hydrogen bonding interactions of carbamate linkages within pre-organized vesicle bilayers (Fig. 1). To verify the importance of pre-organization within bilayers of precursor polymersomes before triggered PICsome formation, PEO₄₅-*b*-P(NCMA_{0.55}-CO-DPA_{0.45})₂₉ was subjected to direct decaging upon UV irradiation in DMSO, followed by self-assembly in aqueous media. We could only observe the formation of micellar nanoparticles instead of polymeric vesicles (Supplementary Fig. 32); note that similar results were obtained for PEO₄₅-*b*-PNCMA₁₇-*b*-PDPA₂₁ triblock copolymer (Supplementary Fig. 35) (page 12, first paragraph; page 16, second paragraph).

Q3. Minor comments to the possible impact of the concept is that the vesicles that are investigated are more than half a micron big. That is generally considered too large to be efficient for most drug delivery applications. I am sure there are possibilities to resize the vesicles that that could be investigated to remedy this. Ideally that is done, however, it would be possible to just comment on this and how it would be addressed.

The Reviewer is absolutely correct concerning the size of vesicles considering their future applications as drug nanocarriers. There exist quite a few techniques to fabricate small-sized polymersomes. For example, direct extrusion of polymersome solution at intermediate levels of water/organic solvent mixture through syringe filters with pre-determined sizes (Liu, *et al. J. Am. Chem. Soc.* **2016**, *138*, 33, 10452); sonication of polymersome dispersion with organic solvent as plasticizing agent (Wilson, *et al. Polym. Chem.* **2016**, *7*, 3977-3982). The current two-step approach, polymersome-to-PICsome transition,

provides a convenient route to regulate PICsome dimensions through controlled manipulation of precursor polymersome sizes.

Q4. Additionally, several compounds are encapsulated and their long-term release profiles are investigated. Here, the release rate is more interesting than the release fraction, which in many cases were quite similar between the block copolymersomes and the PICsomes. The authors also present that data only in relative release. A description of the absolute amounts that could be incorporated into the vesicles should be included for relevance to drug delivery applications.

We conducted additional experiments on this aspect. The drug/model drug loading efficiency, loading content, and loaded drug concentration in $\text{PEO}_{45}\text{-}b\text{-P}(\text{NCMA}_{0.55}\text{-}co\text{-DPA}_{0.45})_{29}$ polymersomes, $\text{PEO}_{45}\text{-}b\text{-PNCMA}_{17}\text{-}b\text{-PDPA}_{21}$ polymersomes, and $\text{PEO}_{45}\text{-}b\text{-P}(\text{DCMA}_{0.45}\text{-}co\text{-PDPA}_{0.55})_{33}$ polymersomes were quantified by fluorescence (Dox·HCl, coumarin 343, and calcein) and UV-Vis absorbance (5-Fu, gemcitabine hydrochloride) measurements, respectively; relevant results are summarized in Supplementary Table 1. As all drugs and model drugs were encapsulated into aqueous interiors of polymersomes under the same self-assembling conditions, i.e., actuating the self-assembly process via the addition of aqueous solutions of corresponding drugs and model drugs at the same molar concentration, the drug loading efficiency and loaded drug concentration are in the range of ~8-9% and 1.0-1.1 mM, respectively (page 23, last paragraph).

Q5. The TEM images indicate a significant fraction of smaller objects that are not captured by the DLS size distributions that are very narrow. Comment?

The newly included Supplementary Fig. 17 shows the TEM image recorded at an enlarged field view for $\text{PEO}_{45}\text{-}b\text{-P}(\text{NCMA}_{0.55}\text{-}co\text{-DPA}_{0.45})_{29}$ polymersomes, revealing the presence of robust polymersomes with generally uniform polymersomes (~500 nm), along with a few smaller ones. DLS measurements report on intensity-average hydrodynamic diameters, which are more sensitive to large polymersomes, compared to smaller ones. This explains why DLS size distributions are unimodal and quite narrow.

Q6. In which way does Figure 7e show “microstructural destruction”?

Figure 7e should read Figure 7d. At 70 °C with 2.0 M NaCl, PICsomes disassemble into unimers and no reliable DLS signals could be detected (Fig. 7d); scattered light intensities also exhibited abrupt decrease upon heat to 70 °C (Supplementary Fig. 33).

Reviewer #2 (Remarks to the Author):

In this submission, Liu and coauthors describe a general method to convert a polymersome to PICsome in situ using stimuli as a handle. This transformation is accompanied with permeability switching from impermeable hydrophobic membrane to semipermeable molecular size-selective membrane. The permeability of the PICsome can be finely tuned by changing the composition of block copolymer and duration of stimuli treatment. The molecular size-selective permeability was further utilized to selectively release cancer drugs with small size difference. The resulted PICsome shows high stability in broad pH range, strong ionic strength, and high temperature, which is not common in classic

PICsomes. This high stability is attributed to the cooperative effect of hydrogen bonding and ionic pairing. Overall, it is an interesting concept to achieve the semipermeable nanomembrane.

We appreciate positive comments from Reviewer 2.

However, following concerns need to be addressed.

1. The polymersome to PICsome transitions all rely on the assumption that decaging of carboxylate protonates the tertiary amine and consequently forms ionic pairs. However, the characterization on the decaging is not solid enough to prove the transformation. The decaging of 2-nitrobenzyl ester was only followed by monitoring the generation of nitroso benzaldehyde in absorption spectrum. But, there is almost no increase in absorbance at 365 nm after UV irradiation (Supplementary Fig. 17). Photolysis of 2-nitrobenzyl ester is well-known to be non-efficient in literatures, which is also supported by the NMR experiment in Supplementary Fig 18. NMR of UV-treated polymersome should be taken in the organic NMR solvent to quantify the conversion of 2-nitrobenzyl to nitroso benzyl aldehyde. The percentage of remaining caged carboxylate can significantly change the pKa resulted polymers. Without clear quantification of the functional group transformation, later discussion on PICsome properties relying on polymer pKa cannot be trusted.

Similarly, the direct evidences need to be given to support decaging of carboxylate by GSH.

Many thanks for the insightful comments from Reviewer 2. Considering characterization of the process of carboxyl photo-decaging by NMR in organic solvents, please refer to *Point 1* in the above replies to Reviewer 1. The photocleavage extent of 2-nitrobenzyl moieties for polymersomes dispersions is >98% upon 10 min UV irradiation (Supplementary Fig. 19; *page 8, third paragraph*).

We need to clarify that ^1H NMR experiments shown in Supplementary Fig. 21 (*Supplementary Fig. 18 in prior version*) were conducted in D_2O for $\text{PEO}_{45}\text{-}b\text{-P}(\text{NCMA}_{0.55}\text{-}co\text{-DPA}_{0.45})_{29}$ polymersome dispersions before and after UV irradiation. Apparently, the vesicular dispersion initially exhibited a bluish tinge, but changed to grayish yellow upon UV irradiation (insets in Supplementary Fig. 21); meanwhile, in both cases only signals of well-solvated PEO coronas could be detected in ^1H NMR spectra, indicating the presence of colloidal aggregates, i.e., PICsomes with immobile chain segments in bilayers, upon UV-triggered carboxyl decaging. Moreover, direct TEM observations (Fig. 3a,b) after UV irradiation revealed the presence of intact and robust vesicular nanostructures (*page 8, 4th paragraph*).

As for UV-Vis absorption spectra recorded for $\text{PEO}_{45}\text{-}b\text{-P}(\text{NCMA}_{0.55}\text{-}co\text{-DPA}_{0.45})_{29}$ polymersome dispersion during UV light irradiation in aqueous media (Supplementary Fig. 18), the absorbance at 260 nm decreased considerably, whereas the absorbance at 365 nm did not exhibit discernible changes. In colloidal dispersions, UV-Vis absorbance changes are complicated and/or compensated by chromophore encapsulation and scattering issues, which might lead to the disappearance of isobestic point during photolysis (Zhao, et al. *Macromolecules* **2006**, *39*, 4633-4640). However, newly updated ^1H NMR results in DMSO (Supplementary Fig. 19) unambiguously confirmed the highly efficient photocleavage of 2-nitrobenzyl moieties. In addition, the photolysis rate of 2-nitrobenzyl ester is related to UV irradiation intensity and concentration (Zhao, et al. *J. Am. Chem. Soc.* **2011**, *133*, 19714-19717). In the current work, the photolysis was conducted with 365 nm UV light at 8 mW/cm² and NCMA concentration of ~0.12 mM.

The reduction-triggered decaging process was further examined with ^1H NMR, exhibiting a decaging extent of >99% upon treating with 10 mM GSH for 24 h (*page 21, 2nd paragraph*; Supplementary Fig. 40).

2. It doesn't make sense that the PICsome doesn't disassemble in pH2.0 while the corresponding polymersome precursor which is supposed to be more hydrophobic disassembled in pH4.0. In these acidic conditions, the electrostatic interaction should be eliminated in both polymersome and PICsome, and the hydrophobic interaction would dominate the driving force for the assembly. The authors don't give any explanation. Decaged PEO₄₅-b-P(NCMA_{0.55}-co-DPA_{0.45})₂₉ polymer should be used to form self-assembly in pH2.0. Its morphology should be the same as the morphology of PICsome that is formed in situ in the extreme pH. Also, the irradiation condition and NMR of decaged PEO₄₅-b-P(NCMA_{0.55}-co-DPA_{0.45})₂₉ polymer should be provided. It is highly likely that irradiation of intrinsically photo-unstable amine containing polymers may induce side reactions that can potentially crosslink the assembled polymer. The PICsome composition and structure need to be rigorously characterized by NMR and GPC.

Precursor polymersomes readily disassemble into irregular aggregates under acidic media (pH < 4.0) (Fig. 3a,c and Fig. 6a,b). Considering the apparent pK_a of tertiary amine moieties in PEO₄₅-b-P(NCMA_{0.55}-co-DPA_{0.45})₂₉ diblock copolymer (Supplementary Fig. 16), acidic milieu-triggered disassembly could be attributed to protonation of tertiary amines in polymersomes bilayers, i.e., the integrity of vesicular nanostructures could not be maintained with the sole contribution of hydrophobic interactions. On the contrary, for corresponding PICsomes obtained via UV irradiation, the microstructural integrity was maintained even under pH 2.0 (Fig. 3b,d and Fig. 6a,b), which could be ascribed to additional hydrogen bonding interactions associated with protonated carboxyl moieties (Supplementary Fig. 27), cooperative hydrogen bonding and π - π interactions between benzyl carbamate side linkages, together with hydrogen bonding interactions between carboxyl and carbamate moieties; cation- π interactions between protonated tertiary amines and benzyl carbamate side linkages might also contribute to the observed PICsome stability at pH 2.0 (Gong, *et al. Nat. Commun.* **2019**, *10*, 5127-5134). Most importantly, the above noncovalent interactions occur cooperatively and at high local concentrations due to the pre-organized nature within precursor polymersome bilayers. Thus, although electrostatic interactions were largely eliminated at pH 2, vascular integrity could still be maintained (*page 14, first paragraph*).

Furthermore, the self-assembly of decaged PEO₄₅-b-P(NCMA_{0.55}-co-DPA_{0.45})₂₉ directly generated via UV irradiation in DMSO was actuated in acidic aqueous media at pH 2.0, revealing the formation of micellar aggregate with non-uniform size distribution instead of PICsomes (Supplementary Fig. 32d). These results suggested that directly decaged polymer still possesses the tendency of self-assembling under pH 2.0. Meanwhile, the fabrication of PICsomes via *in situ* polymersome transformation revealed the importance of pre-organization within precursor polymersome bilayers before triggered PICsome formation.

As mentioned in the above point 1, for the PICsome dispersion generated via UV irradiation, its lyophilized powder could be directly dissolved in organic solvent such as DMSO-*d*₆, thus excluding the possibility of photo-induced crosslinking involving tertiary amine moieties; ^1H NMR results revealed a

photocleavage extent of >98% upon UV irradiation for 10 min (Supplementary Fig. 19; *page 8, third paragraph*). In addition, PICsomes formed by UV irradiation could disassemble into unimers and no reliable DLS signals could be detected in aqueous media at 70 °C in the presence of 2.0 M NaCl (Fig. 7d and Supplementary Fig. 33), further confirming that nanostructures are stabilized by cooperative noncovalent interactions (Fig. 1), instead of chemical crosslinking and side reactions associated with photo-stability issues of amine-containing polymers.

To exclude possible UV-triggered photoreaction and/or decomposition of tertiary amine moieties in BCPs, we further examined the photostability of both *N,N*-diisopropylethylamine and triethylamine against UV irradiation (0-10 min), revealing essentially no discernible chemical structural changes (Supplementary Figs. 23-24; *page 9, first paragraph*).

Due to that the lyophilized powder of PICsome dispersion could not be dissolved in THF, DMF, and aqueous media at room temperature, thus GPC analysis of PICsomes is not feasible. Please refer to Supplementary Fig. 19 for ¹H NMR results conducted in DMSO-d₆.

3. The fluorescence of naphthalimide and Nile red upon light exposure and different pH need to be done as a control experiment.

The evolution of fluorescence emission of naphthalimide and Nile red upon UV light exposure was recorded, revealing ~4% and ~7% decrease of fluorescence intensity after 10 min UV irradiation (8 mW/cm²), respectively (Supplementary Fig. 30), which are almost negligible compared to the observed large extent of fluorescence emission changes. In addition, pH-dependent emission changes of naphthalimide-labeled polymersomes were also measured as control experiments (Supplementary Fig. 28).

4. It is not fair to use release rate of encapsulated cargos to compare the permeability of the membrane without knowing how much of cargo is loaded in the assembly. To be conclusive, the encapsulation of different cargos needs to be similar.

Many thanks for insightful comments and Reviewer 1 also proposed similar comments. Please refer to Point 4 in the above replies to Reviewer 1 (Supplementary Table 1; *page 23, last paragraph*).

Reviewer #3 (Remarks to the Author):

Polymersomes and polyion complex vesicles (PICsomes) are artificial nanostructures mimicking natural membrane structures consisting of aqueous interiors enclosed by hydrophobic bilayers. However, there is a dilemma between polymersomes and PICsomes in that polymersomes possess enhanced stability and compromised membrane permeability, while PICsomes possess semipermeable membranes but low stability towards physiologically relevant ionic strength and temperature. In this manuscript, the authors proposed a strategy to solve this dilemma by stimuli-triggered polymersome-to-PICsome transition with pre-organized block copolymer vesicles and subsequent de-caging reaction. Contrary to conventional PICsomes, the in situ generated ones are highly stable towards extreme pH range (pH 2-12), high ionic strength and elevated temperature, while maintain semipermeable

properties. Moreover, the permeability of the PICsomes could be tuned by the sequence structure of the block copolymers, which enabled on-demand triggered release functions.

This is an interesting and highly important study, and provides a novel paradigm for polymeric vesicles design leveraging stimuli-responsive properties. The whole manuscript is in precise logic and the conclusions are fully supported by the data. The manuscript should be accepted for publication in this journal.

We appreciate positive comments from Reviewer 3.

The following are some minor points for this manuscript:

1. It is interesting to see the comparison between diblock copolymer and triblock copolymer with similar components (PEO₄₅-b-P(NCMA_{0.55}-co-DPA_{0.45})₂₉ vs. PEO₄₅-b-PNCMA₁₇-b-PDPA₂₁). The triblock copolymer showed faster drug release upon UV stimulation than diblock copolymer due to the swelling behavior. One possible reason for this difference is the relatively deficient in ion pair between DPA and NCMA in the block copolymer. However, there are more DPA in the triblock copolymer (n=21) than that in the diblock copolymer (n=13). Will extra non-pairing DPA be another reason for the swelling? The NCMA and DPA are almost equal in the studied diblock copolymer (0.55/0.45). Will the NCMA/DPA ratio be another parameter for adjusting the release rate of the diblock polymer vesicles?

NCMA/DPA molar fractions of PEO₄₅-b-P(NCMA_{0.55}-co-DPA_{0.45})₂₉ and PEO₄₅-b-PNCMA₁₇-b-PDPA₂₁ are ~0.55/0.45 and ~0.45/0.55, respectively, as determined by ¹H NMR analysis (page 6, Table 1). Although comonomer molar ratios of these two types of BCPs (diblock versus triblock copolymer) were not precisely controlled for direct comparison, the other two BCP samples, namely, PEO₄₅-b-P(NCMA_{0.49}-co-DEA_{0.51})₃₂ (0.49/0.51, NCMA/DEA) and PEO₄₅-b-P(DCMA_{0.45}-co-DPA_{0.55})₃₃ (0.45/0.55, DCMA/DPA), could be utilized to elucidate the comonomer molar ratio issue (pages 5-6; Fig. 2 and Table 1).

For PEO₄₅-b-P(NCMA_{0.49}-co-DEA_{0.51})₃₂ diblock copolymer, upon decaging, carboxyl and tertiary amine moieties are close to be equal; however, both DLS and TEM characterization results confirmed similar UV light-triggered polymersome-to-PICsome transition process, as compared to PEO₄₅-b-P(NCMA_{0.55}-co-DPA_{0.45})₂₉ polymersomes (Fig. 3 a,b; Fig. 4 and Supplementary Fig. 22; page 9, first paragraph). Thus, for the diblock copolymer containing randomly copolymerized tertiary amine comonomer and photo-caged monomer, slight deviation from 0.5/0.5 comonomer ratio will not considerably affect the extent of swelling during polymersome-to-PICsome transition.

For GSH-responsive PEO₄₅-b-P(DCMA_{0.45}-co-DPA_{0.55})₃₃ diblock copolymer, the comonomer molar ratio of caged carboxyl to tertiary amine moieties (DCMA/DPA, 0.45/0.55) is quite similar that of PEO₄₅-b-PNCMA₁₇-b-PDPA₂₁ triblock copolymer (NCMA/DPA, 0.45/0.55) (Figs. 1-2). However, <D_h> distributions of PEO₄₅-b-P(DCMA_{0.45}-co-DPA_{0.55})₃₃ polymersomes remained almost unchanged upon treating with 10 mM GSH (Fig. 10a-d), which is quite different to that exhibited by PEO₄₅-b-PNCMA₁₇-b-PDPA₂₁ triblock polymersomes. The above results further confirmed that different extents of swelling upon triggered polymersome-to-PICsome transition could be mainly ascribed to sequence structure of the bilayer forming block (random versus block), as comonomers ratios of BCPs used in the current study are in

the range of 0.45-0.55 (*page 17, first paragraph*). However, we speculate that for triblock copolymer polymersomes, the extent of swelling (i.e., permeability and permselectivity) could be facilely tuned by block ratios, as soluble polyion complex will form when oppositely charged polyelectrolytes falls apart from stoichiometric ratios.

2. DOX and some other drugs are sensitive to UV irradiation. Will that affect the release study?

The photo-stability of drug and model drugs used in the current study was characterized by fluorescence (Dox·HCl, coumarin 343, and calcein) and UV-vis absorbance (5-Fu, gemcitabine hydrochloride) measurements, respectively. All drugs and model drugs exhibited <5% photo-degradation (except for calcein dye, ~12% photo-degradation) upon UV irradiation for 10 min (365 nm, 8 mW/cm²). Thus, the photo-bleaching of these drugs and model drugs will not considerably affect the drug release studies; note that release profiles are reported on the normalized level, and the sustained release duration (20-30 h) is much larger than the irradiation duration (10 min) (*Supplementary Fig. 41; page 24, first paragraph*).

3. The authors used “w/o” for “without”, but “w/o” is easy confusing with “oil-in-water”.

Through the main text, “w/o” and “w/” were replaced with “without” and “with”, respectively (*Fig. 5, Fig. 6, Fig. 7, Fig. 9, and Fig. 10; Supplementary Figs. 21, 22, 34, 36, 37, 38, 39*).

4. In supplementary Fig.17, what does the blue and red arrow represent?

In *Supplementary Fig. 18 (Supplementary Fig. 17 in prior version)*, enlarged spectra in the range of 235-290 nm was provided as the inset; The process of photocleavage was further examined with ¹H NMR in organic solvent (*Supplementary Fig. 19; page 8, third paragraph*).

5. In supplementary Fig.18, the authors used HDO represent the solvent in the NMR spectrum, there may also be H₂O. Simply use “solvent” may be more accurate.

The phrase “HDO” was replaced with “solvent” in *Supplementary Fig. 21 (Supplementary Fig. 18 in prior version)*.

6. Line147, Supplementary “Fig1b” should be “Fig1c”.

Many thanks and all minor mistakes were corrected in the revised version. The entire main text and Supplementary text were thoroughly checked and polished.

Reviewers' Comments:

Reviewer #1:

Remarks to the Author:

The authors have performed almost all the requested corrections and additions from the original report. It is a pity that comparable PICsomes without the pre-assembly and decaging could not be made, as PICsomes without pre-assembly would have been a more elucidating control to answer the question regarding the cause of the stability, as well as a proper control for the release experiments.

I don't agree with all the arguments that the authors do in comparing previous literature to the results in this manuscript. The authors mix thermodynamic arguments about stability with the obvious observation that the structure is kinetically trapped since it could not be formed other than by the two-step process. These lines of arguments are difficult to reconcile, however logical they might sound.

Furthermore, I don't think that all presented conclusions regarding the nature of the bonding can be fully drawn by comparing such different systems under different circumstances. That pH and ionic strength affect charge interactions and hydrogen bonding is, of course, true. However, the question is which interactions dominate and elad to the unique observed result, i.e., why and how. To their credit, the authors present all this as a discussion, which is fair, although I think it could be done in a better and more consistent way if the authors had spent more time on separating their arguments and maybe managed some additional experiment. Of course, we also learn something about the uniqueness of the system from that the PICsomes could not be formed without the two-step method. I think this is fascinating, but it definitely looks like an out-of-equilibrium process to me.

The only remaining issue that requires fixing concerns the figures. The figures, as presented in the manuscript now, do not have proper labels and units. In essence, they are not readable. Please make sure that the correct labels and units are given everywhere before final acceptance. Given the completion of the figures, I think the manuscript can be accepted for publication in Nature Communications.

Reviewer #2:

Remarks to the Author:

The authors have addressed all my concerns satisfactorily. This manuscript is now publishable in Nature Communications.

Reviewer #3:

Remarks to the Author:

The authors have addressed all my concerns. I have no more questions for this manuscript.

RESPONSES TO REFEREES (NCOMMS-19-32638A):

(Reviewer comments in black, our response in blue)

Reviewer #1 (Remarks to the Author):

The authors have performed almost all the requested corrections and additions from the original report. It is a pity that comparable PICsomes without the pre-assembly and decaging could not be made, as PICsomes without pre-assembly would have been a more elucidating control to answer the question regarding the cause of the stability, as well as a proper control for the release experiments. I don't agree with all the arguments that the authors do in comparing previous literature to the results in this manuscript. The authors mix thermodynamic arguments about stability with the obvious observation that the structure is kinetically trapped since it could not be formed other than by the two-step process. These lines of arguments are difficult to reconcile, however logical they might sound.

Thanks for the reviewer's positive comments. We totally agree that PICsomes without pre-assembly would be an excellent control to elucidate underlying mechanisms of nanostructure stability and molecular size-selective bilayer permeability. Using the direct self-assembly approach based on decaged block copolymers, we could only obtain polyion complex micelles instead of PICsomes. Note that polyion complex micelles obtained via direct self-assembly are sensitive to externally added salt (Fig. 7 and Supplementary Fig. 32 and Fig. 35).

We agree with the reviewer that PICsomes generated via two-step pre-assembly approach are kinetically trapped nanostructures rather than thermodynamically stable ones, as PICsome will not form again if they are disintegrated under a combination of elevated temperature and high ionic strength (Fig. 7). However, it should be noted that pre-organization within bilayers of precursor polymersomes before triggered polymersome-to-PICsome transition is crucial for the microstructural stability of resultant PICsomes.

Furthermore, I don't think that all presented conclusions regarding the nature of the bonding can be fully drawn by comparing such different systems under different circumstances. That pH and ionic strength affect charge interactions and hydrogen bonding is, of course, true. However, the question is which interactions dominate and elad to the unique observed result, i.e., why and how. To their credit, the authors present all this as a discussion, which is fair, although I think it could be done in a better and more consistent way if the authors had spent more time on separating their arguments and maybe managed some additional experiment. Of course, we also learn something about the uniqueness of the system from that the PICsomes could not be formed without the two-step method. I think this is fascinating, but it definitely looks like an out-of-equilibrium process to me.

The Reviewer is absolutely correct and many thanks for insightful comments concerning the underlying mechanism of observed PICsome stability. In the current system, multiple noncovalent interactions including hydrogen bonding, π - π , ion-pair, and hydrophobic interactions play their roles in stabilizing PICsome nanostructures. We tentatively speculate that these different types of noncovalent interactions cooperatively contribute to the observed microstructural stability. Due to the pre-organization nature within precursor polymersomes, local densities of functional moieties involved

with these noncovalent interactions are $> 1-2 \text{ M}$ within vesicle bilayers.

The only remaining issue that requires fixing concerns the figures. The figures, as presented in the manuscript now, do not have proper labels and units. In essence, they are not readable. Please make sure that the correct labels and units are given everywhere before final acceptance. Given the completion of the figures, I think the manuscript can be accepted for publication in Nature Communications.

Many thanks and all issues concerning Figure labels and units were solved in the revised version. These labeling issues were generated during word-to-pdf converting. The entire main text and Supplementary text were thoroughly checked.

Reviewer #2 (Remarks to the Author):

The authors have addressed all my concerns satisfactorily. This manuscript is now publishable in Nature Communications.

We appreciate positive comments from Reviewer 2.

Reviewer #3 (Remarks to the Author):

The authors have addressed all my concerns. I have no more questions for this manuscript.

We appreciate positive comments from Reviewer 3.